# OpenLex3D: A Tiered Evaluation Benchmark for Open-Vocabulary 3D Scene Representations

**Christina Kassab**[*1]    **Sacha Morin**[*2,4]    **Martin Büchner**[*3]    **Matías Mattamala**[1]

**Kumaraditya Gupta**[2,4]    **Abhinav Valada**[3]    **Liam Paull**[2,4,5]    **Maurice Fallon**[1]

[1]University of Oxford    [2] Université de Montréal    [3]University of Freiburg
[4] Mila - Quebec AI Institute    [5]Canada CIFAR AI Chair

## Abstract

3D scene understanding has been transformed by open-vocabulary language models that enable interaction via natural language. However, at present the evaluation of these representations is limited to datasets with closed-set semantics that do not capture the richness of language. This work presents OpenLex3D, a dedicated benchmark for evaluating 3D open-vocabulary scene representations. OpenLex3D provides entirely new label annotations for scenes from Replica, ScanNet++, and HM3D, which capture real-world linguistic variability by introducing synonymical object categories and additional nuanced descriptions. Our label sets provide 13 times more labels per scene than the original datasets. By introducing an open-set 3D semantic segmentation task and an object retrieval task, we evaluate various existing 3D open-vocabulary methods on OpenLex3D, showcasing failure cases, and avenues for improvement. Our experiments provide insights on feature precision, segmentation, and downstream capabilities. The benchmark is publicly available at: https://openlex3d.github.io.

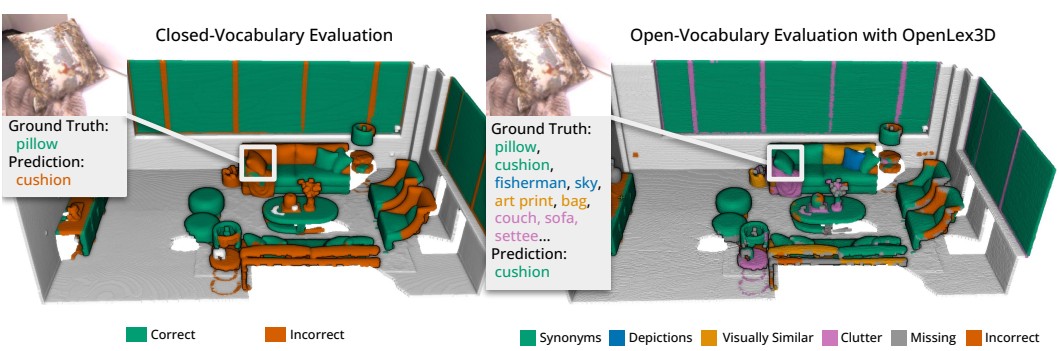

Figure 1: **The OpenLex3D evaluation benchmark enables more detailed analysis of open-vocabulary 3D scene representations than closed-vocabulary evaluation methods.** We compare the same open-vocabulary representation when assessed under closed-vocabulary semantics (left) and using OpenLex3D labels (right). In contrast to closed-vocabulary methods where a prediction must match the exact ground truth label, OpenLex3D provides a manifold of label categories of varying precision: *synonyms* being the most precise; *depictions*, which include, e.g., printed images on objects; *visually similar*, which refer to objects with comparable appearance; and *clutter*, which accounts for label perturbation due to imprecise segmentation.

39th Conference on Neural Information Processing Systems (NeurIPS 2025) Track on Datasets and Benchmarks.

# 1  Introduction

3D scene understanding is a key capability enabling embodied agents to perceive, interpret, and interact with the physical world. An effective scene representation should generalize across diverse indoor and outdoor environments [3]. The introduction of visual-language models such as CLIP [25] and LLaVa [18] has transformed this field, allowing objects to be labeled using free-form semantics rather than being constrained to a closed-set of object categories. This has motivated the incorporation of open-vocabulary vision-language models in 3D scene understanding [6, 9, 13, 15, 20, 24, 32, 35]

Evaluating closed-set semantics is relatively straightforward—the predicted class label of each point-wise prediction either matches the point's ground truth label or does not — as shown in Fig. 1. In contrast, assessing the performance of open-vocabulary models is more challenging and is not yet well defined by a benchmark. Published works on open-vocabulary representations [32, 35] have typically used closed-set semantic segmentation labels and metrics—despite this underlying mismatch. This defeats the purpose and flexibility of open-vocabulary predictions by constraining the model assessment to a limited set of evaluation labels [23, 32].

We argue that relying on closed-set evaluation overlooks the nuance of real-world language labeling, which is rarely constrained to a single label per object. For example, a couch might also be referred to as a "sofa" or "seating". Prior work has proposed using existing ontologies like WordNet (a large English language lexical database) [22] to mitigate these ambiguities in language benchmarks, though differentiating between *similarities* and *associations* remains an open question [8]. Several methods have sought to evaluate downstream performance by focusing on tasks such as visual question answering and object retrieval [4, 5, 38]. However, this only provides sparse estimates of the overall fidelity of the underlying scene representation.

In this work, we aim to overcome these limitations by introducing OpenLex3D, a novel benchmark for evaluating open-vocabulary scene representation methods. OpenLex3D introduces *four different label categories* of description precision: *synonyms*, *depictions*, *visual similarity*, and *clutter*. We use the categories to evaluate the performance of a method in capturing the correct labels (*synonyms*) while also diagnosing different types of misclassification. Our benchmark is implemented by relabeling scenes from three widely used indoor RGB-D datasets with a new set of human-annotated ground truth labels. Our label sets contain between 300 and 1200 unique labels per scene and have *13 times more labels per scene* than the original datasets. To summarize, we make the following contributions:

1. We introduce a new labeling scheme where each object is described by multiple free-form text labels organized into four categories of different linguistic levels of precision.

2. We provide OpenLex3D labels for a total of 3812 objects from Replica [31], Scannet++ [39] and Habitat-Matterport 3D [26]. Each object has been labeled by four human annotators.

3. We propose two evaluation tasks building on the OpenLex3D labels: tiered semantic segmentation and object retrieval—including newly proposed metrics reflecting performance at the different precision levels introduced. Our prompt lists for segmentation contain up to 3500 labels per dataset and our retrieval query set contains up to 1500 queries per scene (see Tab. 1), enabling comprehensive evaluation of several state-of-the-art 3D open-vocabulary methods.

4. We make the OpenLex3D toolkit and ground truth data publicly available at: https://openlex3d.github.io.

# 2  Related Work

**Open-Vocabulary Scene Representations:** The recent development of Visual-Language Models (VLMs) has motivated their integration in both *object-centric* and *dense* map representations:

*Object-centric representations* explicitly factorize scene geometry as a set of 3D objects and represent semantic information as object-level open-vocabulary features from vision-language encoders such as CLIP [25]. Methods typically derive object features by fusing features from multiple views using various strategies [12]. This makes them a compact representation for embodied AI applications that involve object-level understanding and interaction, such as object retrieval. Methods such as OpenMask3D [32] and OpenIns3D [10] first determine candidate objects using instance segmentation

| Dataset | Scene Name | No. of Objects | No. of Unique Orig. Labels | No. of Unique OL3D Labels | No. of Labels per Object (Avg) | No. of Labels per Object (Max) | No. of Ambiguous Objects | Sem. Seg. Prompt List Size | No. of *S* Queries | No. of *S+D* Queries | Total Number of Queries |
|---|---|---|---|---|---|---|---|---|---|---|---|
| Replica | room0 | 92 | 28 | 478 | 16 | 33 | 6 | 1150 | 236 | 373 | 609 |
| | room1 | 52 | 24 | 297 | 14 | 26 | 0 | 1150 | 164 | 93 | 257 |
| | room2 | 61 | 21 | 336 | 15 | 27 | 1 | 1150 | 165 | 122 | 287 |
| | office0 | 57 | 24 | 352 | 13 | 25 | 7 | 1150 | 195 | 211 | 406 |
| | office1 | 43 | 22 | 292 | 11 | 30 | 4 | 1150 | 163 | 70 | 233 |
| | office2 | 68 | 21 | 391 | 14 | 30 | 3 | 1150 | 208 | 87 | 295 |
| | office3 | 83 | 26 | 409 | 14 | 27 | 2 | 1150 | 211 | 112 | 323 |
| | office4 | 55 | 16 | 330 | 14 | 28 | 2 | 1150 | 176 | 122 | 298 |
| ScanNet++ | 49a82360aa | 127 | 48 | 707 | 13 | 32 | 1 | 3407 | 433 | 409 | 842 |
| | 1f7cbbdde1 | 154 | 55 | 1187 | 14 | 35 | 0 | 3407 | 629 | 782 | 1411 |
| | 8a35ef3cfe | 97 | 43 | 575 | 11 | 30 | 1 | 3407 | 303 | 532 | 835 |
| | 0a76e06478 | 110 | 41 | 750 | 13 | 26 | 1 | 3407 | 356 | 1068 | 1424 |
| | 4c5c60fa76 | 295 | 62 | 1110 | 12 | 25 | 0 | 3407 | 633 | 864 | 1497 |
| | 0a7cc12c0e | 134 | 60 | 654 | 8 | 17 | 0 | 3407 | 377 | 169 | 546 |
| | c0f5742640 | 131 | 49 | 730 | 12 | 23 | 0 | 3407 | 414 | 488 | 902 |
| | fd361ab85f | 79 | 36 | 454 | 12 | 30 | 1 | 3407 | 250 | 137 | 387 |
| HM3D | 000824 | 307 | 72 | 404 | 5 | 12 | 6 | 2350 | 340 | 173 | 513 |
| | 000829 | 170 | 71 | 500 | 8 | 26 | 8 | 2350 | 372 | 293 | 665 |
| | 000843 | 234 | 67 | 514 | 7 | 19 | 63 | 2350 | 367 | 98 | 465 |
| | 000847 | 306 | 69 | 521 | 6 | 18 | 22 | 2350 | 424 | 128 | 552 |
| | 000873 | 345 | 102 | 935 | 11 | 38 | 39 | 2350 | 623 | 945 | 1568 |
| | 000877 | 345 | 139 | 831 | 10 | 33 | 34 | 2350 | 569 | 518 | 1087 |
| | 000890 | 467 | 107 | 829 | 8 | 22 | 42 | 2350 | 624 | 216 | 840 |

Table 1: **Summary of the OpenLex3D labels, segmentation prompt lists, and object retrieval queries per scene across all categories.** OL3D stands for OpenLex3D, *S* stands for synonyms and *D* for depictions. The total number of labeled objects across all scenes is 3812, with as many as 38 unique labels for an individual object. HM3D has the highest number of difficult-to-classify (ambiguous) objects. Our label sets provide *13 times more labels* than the original datasets per scene. We also provide label distributions in Supp. Sec. E.

and then assign CLIP embeddings to each object instance. A similar approach has been followed by open-vocabulary scene graph methods, which use the object vocabulary instances as graph nodes, and then extend the nodes with edges that encode relationships or affordances, which can also be obtained by means of a VLM such as LLaVa [18] or GPT-4V [37]. Open-vocabulary scene graphs include ConceptGraphs [6], HOV-SG [35], Clio [21], and Open3DSG [15].

*Dense representations* instead yield dense open-vocabulary feature embeddings for every point in a scene. The underlying geometric representations include voxels and point clouds, or more recently, neural fields or Gaussian splats. These methods include OpenScene [23], ConceptFusion [11] and SAS [17] and produce dense point cloud maps, where each point stores an open-vocabulary embedding. More recent work uses radiance fields to encode open-vocabulary features [13] [33] [30] [19]. This enables the rendering of photo-realistic novel views with pixel-wise open-vocabulary feature labeling. Similar ideas have also been used with 3D Gaussian representations [24] [7] [36].

**Evaluating Open-Vocabulary Representations:** Recent works focus on two tasks to evaluate open-vocabulary 3D representations: semantic segmentation and object retrieval given a text query.

*Semantic Segmentation:* Open-vocabulary 3D representations can achieve class prediction by comparing their features with a *prompt list* to output the top-ranking class for every point, and then rely on conventional segmentation metrics for evaluation. Common datasets include ScanNet++ (100 classes for evaluation) [39] and Replica (101 classes) [31]. More recent benchmarks such as OpenScan [40], expand the original label sets to include attributes such as property or type or provide per-mask captions such as Mosiac3D [16]. Most of these attributes are derived using an NLP knowledge graph. Additionally, these approaches do not take into account the nuance of real-world object labeling, where multiple correct labels are possible.

*Object Retrieval:* An object-centric representation can serve as a retrieval system by returning objects sharing similar features with a user-specified text query. We treat retrieval as an instance segmentation task, while another line of work focuses on bounding-box regression [2, 34]. The flexibility of the task makes it challenging to automate evaluation, and previous query results do not cover all scene objects. ConceptGraphs [6] evaluates recall for 60 queries, while OpenScene [23] visually verifies retrieval results for 11 classes. Automated evaluation is possible when narrowing the scope of the queries to class labels [20, 32] at the cost of query diversity. The first OpenSun3D [5] workshop challenge provides replicable retrieval evaluation on multiple scenes but is restricted to 25 queries.

In contrast, OpenLex3D overcomes these challenges by providing an extensive set of human-annotated labels for each object. These labels account for natural variability in language and are categorized in terms of description specificity. We complement our benchmark with two novel segmentation metrics to explore both the top predictions and the complete ranking of the prompt list. In addition, we leverage the OpenLex3D category labels to procedurally generate hundreds of queries per scene as

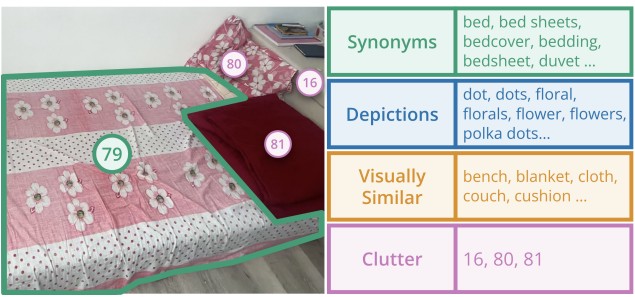

Figure 2: **OpenLex3D label example on ScanNet++ [39]**. We provide not only synonyms for the object but also labels for various potential failure cases, including depictions visible on the target object (e.g., flower prints), visually similar objects (e.g., blanket), and surrounding clutter (indicated by the IDs of the neighboring objects).

part of an automated benchmark. Our queries cover all scene objects and include specific information such as motifs, cultural references, and brands.

## 3 The OpenLex3D Benchmark

Our benchmark consists of multiple ground truth labels for each object that are categorized into different levels of specificity. Those category levels are arranged hierarchically and thus allow us to identify potential classification failure cases and to analyze why they occur. The tiered category approach is inspired by SimLex-999 [8], which attempted to assign more accurate similarity scores between word pairs by carefully distinguishing between *similarities* and *associations* or *relatedness*. The four categories we consider, in decreasing order of description specificity, are:

**Synonyms:** This category includes the primary labels for the target object as well as any other equally valid label. For instance, "glasses" and "spectacles".

**Depictions:** Labels in this category describe any images or patterns depicted on the target object. For example, if a pillow features an image of a tree, the label "tree" would fall under this category.

**Visually Similar:** This includes objects that appear to be visually similar to the target objects and are likely to be confused for it. For example, visually similar terms for "glasses" could include "sunglasses" or "goggles".

**Clutter:** This category covers nearby or surrounding objects. Surrounding object features may "leak" into the features of interest due to 1) co-visibility in the same RGB frames and/or 2) incorrect merging in object-centric representations. This is the only category not defined by text labels but by object IDs pointing to neighboring objects in 3D.

We visualize an example of these categories in Fig. 2. An ideal 3D scene representation would generate high-scoring predictions that fall exclusively under the *synonym* category. However, in practice, most representations yield predictions distributed across all categories, therefore providing insights into different failure modes. For example, a representation yielding many predictions in the *clutter* category may indicate erroneous under-segmentation or merging of objects such as small items on tables. Conversely, a representation with many predictions in the *visually similar* category might point to issues with the open-vocabulary model itself or the feature-merging strategy used, suggesting a failure to differentiate between visually similar but non-synonymical objects.

We employ nouns (including multiple word labels such as "sofa cushion") for our ground truth labels to reduce the ambiguities of sentences and captions. For instance, sentence embedding models [27] are sensitive to variations in word order and struggle to distinguish between sentences with similar structures but different meanings. By using diverse nouns, we can still capture semantic similarities without introducing such challenges.

### 3.1 Label Acquisition and Processing

To implement the benchmark, we create an entirely new set of semantic labels for scenes from three prominent RGB-D datasets (see Supp. Sec. C for additional details):

**Replica** [31] consists of a set of high-quality reconstructions of indoor spaces, including offices, bedrooms, and living spaces. We use eight scenes (*room0-room2* and *office0-office4*) and utilize the camera trajectories provided by NICE-SLAM [41].

**ScanNet++** [39] is a large-scale dataset of a variety of real indoor scenes. Each scene provides laser scanner-registered images from a DSLR camera and RGB-D streams from an iPhone. We use the RGB-D iPhone images, along with their associated poses (metrically-scaled COLMAP [29] trajectories) from eight scenes.

**HM3D** [26] consists of high-resolution 3D scans of building-scale residential, commercial, and civic spaces. We use the trajectories generated by HOV-SG [35] and obtain RGB-D images as well as semantic ground truth across seven scenes of the *Habitat-Matterport 3D Semantics* split.

The goal of *OpenLex3D* is to provide a set of high-quality human-annotated labels for evaluation on commonly used scenes, rather than prioritizing a large quantity of scenes for training purposes.

### 3.1.1 Annotation Process

To generate the labeling data, we first obtained a set of representative images for each ground truth object instance using a method similar to that described in OpenMask3D [32]. For each object instance in the ground truth point cloud, we re-projected its 3D points to the camera plane for labeling. These projected 2D points were used to define a bounding box that encloses the object in the image. Additionally, we used the projected points to generate a Segment Anything (SAM) [14] mask, which provided guidance for the annotator on the object to be labeled.

The annotators were required to fill in responses for the first three categories only (*synonyms*, *depictions*, and *visually similar*), and each object in every scene was labeled by four different annotators. The full instructions given to the annotators are included in Supp. Sec. D. In addition, we illustrate the annotation interface in Supp. Fig. 7. The *clutter* category was filled in post-processing after the annotation process (see Sec. 3.1.2).

We recruited annotators of varying backgrounds and professions from four different countries. As such, we opted for direct recruitment instead of relying on third-party platforms such as Amazon Mechanical Turk (MTurk), which was shown to yield a significant number of incorrect labels [28].

### 3.1.2 Label Post-processing

After obtaining the labels, we curated them to correct spelling errors and eliminate invalid responses. The reviewing process followed these conventions: 1. If the same label occurred in multiple categories, it was assigned to the lowest (least specific) category. 2. If there was significant disagreement in the synonyms category for the target object, the object was kept but flagged as "ambiguous" to indicate that it was difficult to recognize. 3. Variations such as plurals and differences in spacing were permitted and remained unaltered (e.g., "shelves" and "shelf" or "counter top" and "countertop". Regarding the *clutter* category, we identified nearby or surrounding objects based on their 3D Intersection over Union (IoU). We used the Objectron [1] implementation to estimate bounding boxes and compute the IoU. Any neighboring objects with an IoU greater than 0 with the target object were assigned to the *clutter* category.

## 3.2 Evaluation Methodology

We propose two tasks to evaluate open-vocabulary 3D scene representation methods: a semantic segmentation task and an object retrieval task.

### 3.2.1 Task 1: Tiered Open-Set Semantic Segmentation

In the first task, we evaluate the point-level open-vocabulary features stored in a representation using our category-based ground truth. For every 3D point $p$ in the ground truth, benchmarked methods must predict a language-aligned feature vector that can be compared with the label features of a prompt list $\mathcal{L}$ using the cosine similarity, yielding a ranked list $\mathcal{L}^p$ per point $p$.

We use separate prompt lists for each dataset. Each prompt list consists of the unique labels across all categories, objects, and scenes from a given dataset, and contains between 1,000 and 3,000 unique

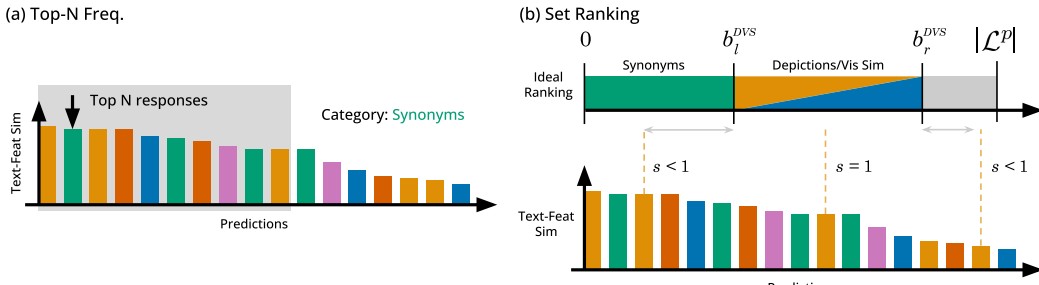

Figure 3: **Top-N Frequency and Set Ranking metrics illustration**. (a) Top-N Freq. measures whether any of the top-N responses contain a label from category C. (b) Set Ranking evaluates the ranking of responses, assessing how closely the predicted rankings align with ideal rankings of categories.

labels each (Tab. 1). This mitigates the problem of positive bias toward the correct labels [12]. To evaluate the performance of a method, we introduce two metrics: Top-N Frequency and Set Ranking.

**Top-N Frequency at Category:** In our setting, each ground truth object is assigned multiple and unique labels. This makes direct application of standard mIoU infeasible (for more details see Supp. Sec. B). To address this, we define a new metric that characterizes the proportion of 3D points that are classified into a given category $C$ within a scene. Our metric is defined as

$$\mathcal{F}_N^C = \frac{1}{|\mathcal{O}|} \sum_{o \in \mathcal{O}} \frac{1}{n_o} \sum_{p \in o} \mathbf{1}(\hat{C}(\mathcal{L}_{1:N}^p) = C), \tag{1}$$

where $\mathbf{1}(\cdot)$ is the indicator function. We normalize by the number of points $n_o$ in any given ground truth object $o \in \mathcal{O}$ to ensure that the metric is not skewed by objects with a large number of points. $\hat{C}$ assigns a category based on $\mathcal{L}_{1:N}^p$ — the top-N most similar labels in $\mathcal{L}^p$ (Fig. 3 a))— and the OpenLex3D categories for $p$. This helps to mitigate any inconsistencies in the ground truth labels (Supp. Sec. H.1). Specifically, $\hat{C}$ predicts *synonym* if $\mathcal{L}_{1:N}^p$ includes a synonym, *depictions* if $\mathcal{L}_{1:N}^p$ includes a depiction but no synonyms and *visually similar* if $\mathcal{L}_{1:N}^p$ includes a visually similar label but no synonyms or depictions. Absent the first three categories, $\hat{C}$ predicts *clutter* if $\mathcal{L}_{1:N}^p$ includes a label that falls into any category of a neighboring object. If a method omits to predict some features for a ground truth point, $\hat{C}$ returns *missing*.

**Set Ranking:** Our second metric assesses the distribution of the label-feature similarity of each point in the scene representation. For this, we quantify the mismatch of the responses when compared against an *ideal tiered ranking* of category sets. We establish *synonyms* ($S$) as the first-rank set, while *depictions* and *visually similar* are considered as a joint second-rank set ($DVS$). Within both sets, we do not assume any label ordering as each label within a category set $y \in \mathcal{Y}_C^p$ is equally explanatory. The size of the sets is determined by the number of corresponding labels in the ground truth categories for each point as shown in Fig. 3(b). We only consider matched pairs of points between ground truth points $p \in \mathcal{P}$ and predicted points, yielding $\mathcal{P}_*$, thus not evaluating missing or falsely predicted points. Our proposed metric is computed for each point $p \in \mathcal{P}_*$ and its ranked predictions $\mathcal{L}^p$ in the predicted point cloud $\mathcal{P}_*$. We sort the ground truth labels according to the ideal set ranking (*synonyms* first, then *depictions* and *visually similar*) and obtain left and right ranking bounds for each set, $b_l^C$ and $b_r^C$, where $C$ denotes the category type ($S$ or $DVS$). We then compute a *rank score* $s_i$ for each prediction $i$, as a function of its rank $r_i$ compared to the ground truth bounds:

$$s(r_i) = \min\left(1 + \min\left(0, \frac{r_i - b_l^C}{b_l^C}\right), 1 - \max\left(0, \frac{r_i - b_r^C}{|\mathcal{L}^p| - b_r^C}\right)\right). \tag{2}$$

If the prediction falls in the correct category set, we obtain $s(r_i) = 1.0$, and $s(r_i) < 1.0$ otherwise. The rank scores are then used to determine the set inlier rates $R_S$ and $R_{DVS}$ as:

$$R_C = \frac{1}{|\mathcal{P}_*|} \sum_{p \in \mathcal{P}_*} \left( \frac{1}{|\mathcal{Y}_C^p|} \sum_i^{|\mathcal{Y}_C^p|} \mathbf{1}\left(s(r_i) = 1\right) \right), \tag{3}$$

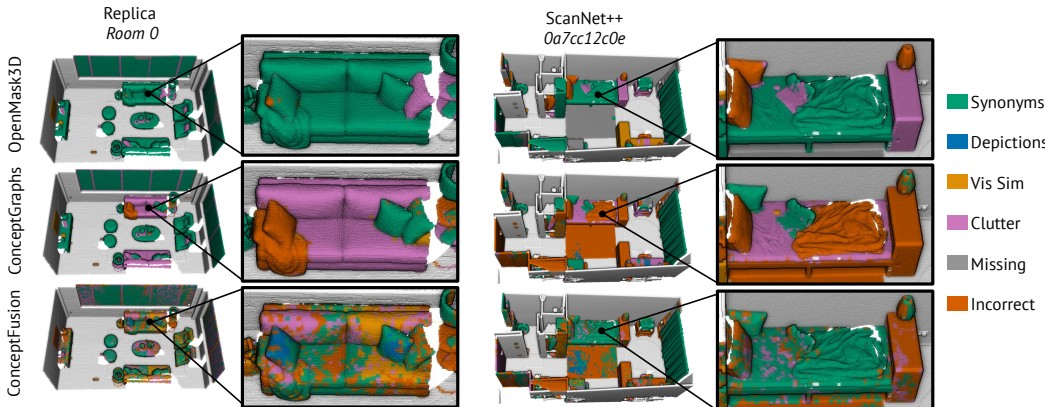

Figure 4: **Top-5 Freq. results for category classification for OpenMask3D [32], ConceptGraphs [6] and ConceptFusion [11] colored by category class.** Object-centric methods that segment in 3D, like OpenMask3D (top), often miss points due to generalization or depth quality issues. Those merging 2D segments tend to merge smaller ones, leading to misclassifications (middle). Dense representations, such as ConceptFusion, produce noisier predictions due to point-level features aggregating information from various context scales. In the highly cluttered environments of ScanNet++ [39], all evaluated methods show reduced performance.

where $|\mathcal{Y}_C^p|$ denotes the number of labels per point $p \in \mathcal{P}_*$ for the particular set $C$ ($S$ or $DVS$). In addition, we compute penalty scores that constitute inverse set ranking scores quantifying underscoring of *synonyms* and both under- and overscoring of the labels within the $DVS$ category set. Those are defined as $P_S^{\leftharpoondown}$, $P_{DVS}^{\leftharpoondown}$, and $P_{DVS}^{\rightharpoonup}$. As formalized in Suppl. Sec. F, they quantify the degree to which the score distribution falls short of satisfying the underlying box constraints and thus go from zero (low penalty) to one (high penalty). Lastly, we also report a mean ranking score $mR$, defined as:

$$mR = \frac{1}{|\mathcal{P}_*|} \sum_{p \in \mathcal{P}_*} \left( \frac{1}{|\mathcal{Y}_{S+DVS}^p|} \sum_{i}^{|\mathcal{Y}_{S+DVS}^p|} s(r_i) \right), \tag{4}$$

where $|\mathcal{Y}_{DVS+S}^p|$ is the number of elements in the ground truth sets $S$ and $DVS$ for each point $p$.

### 3.2.2 Task 2: Open-Set Object Retrieval

The object retrieval task involves segmenting object instances that correspond to a given text-based query in a similar manner to the OpenSun3D Challenge [5]. Here, methods must predict a set of objects, each described with a 3D point cloud and an object-level language-aligned feature vector. For query generation, we use the synonyms in the OpenLex3D label set, along with combinations of synonyms and their associated depictions using the template "[depiction] [synonym]". The resulting queries include references to motifs ("polka dots duvet cover"), specific characters ("ironman portrait") and brands ("nike athletic sneaker"). The number of queries ranges from 200 to 1,500 per scene. For evaluation, we use the Average Precision (AP). We report $AP_{50}$ (IoU of 50%), $AP_{25}$ (IoU of 25%), and mAP scores averaged over the IoU range of [0.5 : 0.95 : 0.05].

## 4 Experiments

In this section, we present a benchmark evaluation of four state-of-the-art object-centric methods and two dense methods across two tasks: semantic segmentation and object retrieval. Implementation details for each method are provided in Supp. Sec. G. We exclude floors, ceilings, and walls from our evaluation and downsample the point clouds to a resolution of $0.05\,\mathrm{m}$ using voxel-based downsampling. We use the ViT-H-14 CLIP backbone for all methods except OpenScene, which uses the ViT-L OpenSeg backbone.

### 4.1 Open-Set Semantic Segmentation

**Top-N Freq.:** We report the Top-5-frequency $\mathcal{F}_{N=5}$ in Tab. 2. The frequencies at 1 and 10 are provided in Supp. Sec. H.1. Top-down views of a selection of the output point clouds colored by category are presented in Fig. 4. Examples of specific predictions are shown in Fig. 5.

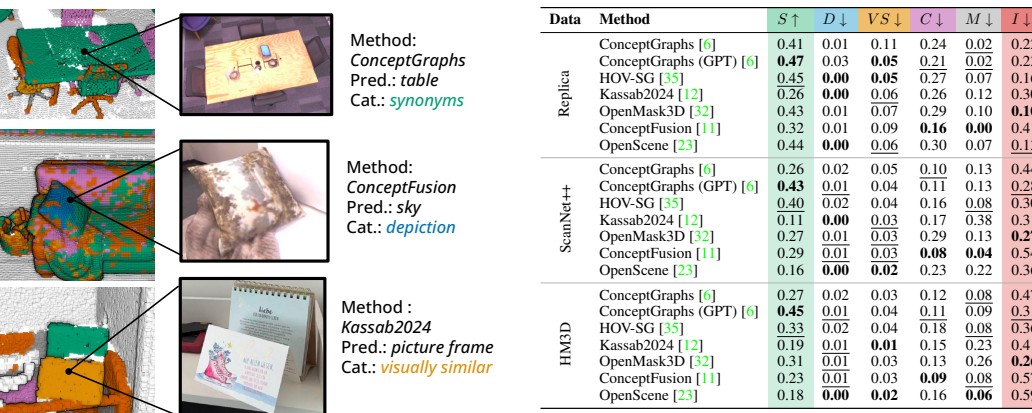

Figure 5: **Example predictions and categories.** We show a correctly predicted label (top). *Depictions* handles cases in which the image depicted on an object is mistaken for the object itself (middle). The *visually similar* category handles reasonable but unprecise predictions (bottom).

Table 2: **Freq. Top 5 Results for Object-Centric and Dense Representations.** *S* is the frequency of synonyms, *D* is depictions, *VS* is visually similar, *C* is clutter, *M* is missing, and *I* is incorrect. Optimal performance would be a $\mathcal{F}^S$ score of 1 while achieving a $\mathcal{F}$ score of 0 for all other categories.

| Data | Method | $S\uparrow$ | $D\downarrow$ | $VS\downarrow$ | $C\downarrow$ | $M\downarrow$ | $I\downarrow$ |
|---|---|---|---|---|---|---|---|
| Replica | ConceptGraphs [6] | 0.41 | 0.01 | 0.11 | 0.24 | 0.02 | 0.22 |
| | ConceptGraphs (GPT) [6] | **0.47** | 0.03 | **0.05** | 0.21 | 0.02 | 0.23 |
| | HOV-SG [35] | 0.45 | **0.00** | 0.05 | 0.27 | 0.07 | 0.16 |
| | Kassab2024 [12] | 0.26 | **0.00** | 0.06 | 0.26 | 0.12 | 0.30 |
| | OpenMask3D [32] | 0.43 | 0.01 | 0.07 | 0.29 | 0.10 | **0.10** |
| | ConceptFusion [11] | 0.32 | 0.01 | 0.09 | **0.16** | **0.00** | 0.41 |
| | OpenScene [23] | 0.44 | **0.00** | 0.06 | 0.30 | 0.07 | 0.13 |
| ScanNet++ | ConceptGraphs [6] | 0.26 | 0.02 | 0.05 | 0.10 | 0.13 | 0.44 |
| | ConceptGraphs (GPT) [6] | **0.43** | 0.01 | 0.04 | 0.11 | 0.13 | 0.28 |
| | HOV-SG [35] | 0.40 | 0.02 | 0.04 | 0.16 | 0.08 | 0.30 |
| | Kassab2024 [12] | 0.11 | **0.00** | 0.03 | 0.17 | 0.38 | 0.31 |
| | OpenMask3D [32] | 0.27 | 0.01 | 0.03 | 0.29 | 0.13 | 0.27 |
| | ConceptFusion [11] | 0.29 | 0.01 | 0.03 | **0.08** | **0.04** | 0.54 |
| | OpenScene [23] | 0.16 | **0.00** | 0.02 | 0.23 | 0.22 | 0.36 |
| HM3D | ConceptGraphs [6] | 0.27 | 0.02 | 0.03 | 0.12 | 0.08 | 0.47 |
| | ConceptGraphs (GPT) [6] | **0.45** | 0.01 | 0.04 | 0.11 | 0.09 | 0.31 |
| | HOV-SG [35] | 0.33 | 0.02 | 0.04 | 0.18 | 0.08 | 0.36 |
| | Kassab2024 [12] | 0.19 | 0.01 | **0.01** | 0.15 | 0.23 | 0.41 |
| | OpenMask3D [32] | 0.31 | 0.01 | 0.03 | 0.13 | 0.26 | **0.26** |
| | ConceptFusion [11] | 0.23 | 0.01 | 0.03 | **0.09** | 0.08 | 0.57 |
| | OpenScene [23] | 0.18 | **0.00** | 0.02 | 0.16 | **0.06** | 0.59 |

As shown in Tab. 2, ConceptGraphs (GPT) is the top-performing method in the *synonyms* category. The GPT prompt generates precise descriptions of the target object, which is then encoded into a highly specific text embedding using the CLIP text encoder. In contrast, the CLIP image encoder, used by other methods, encodes both object-related and broader contextual information from the image, making it more prone to confusion due to visually similar labels or depictions. HOV-SG achieves the next best results. In general, dense methods yield noisier predictions as they use per-pixel features (Fig. 4 and Fig. 5 (middle)).

Regarding *depictions* and *visually similar* categories, OpenScene and Kassab2024 consistently yield the best results. This may stem from their distinct feature association strategies. OpenScene, unlike ConceptFusion, relies solely on per-pixel embeddings without relying on global or region-level merging strategies. Similarly, Kassab2024 selects a feature for each object using Shannon entropy without feature merging. This may preserve feature granularity and reduce classification confusion.

The *clutter* category has worse frequency results across all methods, suggesting that crop scaling and/or segmentation is critical in improving overall classification performance. This is also apparent in the *missing* category. Methods that perform initial segmentation in 3D (Kassab2024 and OpenMask3D) tend to have more missing points compared to those that segment at an image level (ConceptGraphs and HOV-SG). In general, most methods struggle with ScanNet++ and HM3D, indicating that cluttered, real-world environments still pose challenges for all approaches.

**Set Ranking:** In Tab. 3, we report set ranking results. In general, the mean results are high, suggesting that *synonyms* tend to score higher in the predicted ranks, and that *depictions* and *visually similar* labels generally score below *synonyms*, described as the ideal ranking in Sec. 3.2.1. We observe that across all three datasets, HOV-SG [35] and ConceptGraphs [6] yield consistently high mean set ranking scores, while OpenScene [23] and OpenMask3D [32] achieve worse results on ScanNet++.

Furthermore, we observe high $R_S$ scores for ConceptGraphs (GPT), similar to the top-5 frequency $\mathcal{F}^S_{N=5}$, again suggesting that the text embeddings generated from GPT captions are highly specific compared to CLIP image encodings. However, ConceptGraphs (GPT) also consistently overscores the *depictions* and *visually-similar* categories (high $P_{DVS}^{\rightarrow}$) while also underscoring *synonyms* compared to the remaining methods (high $P_S^{\leftarrow}$). This implies that multi-view aggregation of CLIP predictions, as executed by HOV-SG, OpenMask3D, and ConceptGraphs, better approximates the desired set distribution compared to GPT predictions, which yields rather stochastic hits resulting in high $R_S$ scores. As demonstrated, our proposed set ranking evaluation sheds light on non-top-performing scores and quantifies the underlying score distribution compared to the Top-N-frequency metric.

| Data | Method | $mR\uparrow$ | $R_S\uparrow$ | $P_S^{\leftarrow}\downarrow$ | $R_{DVS}\uparrow$ | $P_{DVS}^{\leftarrow}\downarrow$ | $P_{DVS}^{\rightarrow}\downarrow$ |
|---|---|---|---|---|---|---|---|
| Replica | ConceptGraphs [6] | 0.82 | 0.13 | 0.14 | 0.06 | 0.63 | 0.23 |
| | ConceptGraphs (GPT) [6] | 0.63 | **0.21** | 0.33 | **0.07** | 0.52 | 0.43 |
| | HOV-SG [35] | 0.82 | 0.17 | 0.14 | **0.07** | 0.50 | 0.23 |
| | Kassab2024 [12] | 0.76 | 0.10 | 0.21 | 0.03 | 0.54 | 0.27 |
| | OpenMask3D [32] | 0.83 | 0.17 | 0.12 | 0.06 | 0.51 | 0.21 |
| | ConceptFusion [11] | 0.76 | 0.11 | 0.21 | 0.05 | 0.57 | 0.28 |
| | OpenScene [23] | **0.85** | 0.16 | **0.10** | 0.05 | 0.53 | **0.21** |
| ScanNet++ | ConceptGraphs [6] | 0.80 | 0.09 | 0.19 | 0.03 | 0.59 | 0.24 |
| | ConceptGraphs (GPT) [6] | 0.66 | **0.18** | 0.31 | 0.03 | 0.60 | 0.40 |
| | HOV-SG [35] | **0.84** | 0.15 | **0.14** | **0.04** | 0.64 | **0.19** |
| | Kassab2024 [12] | 0.72 | 0.05 | 0.26 | 0.01 | 0.60 | 0.30 |
| | OpenMask3D [32] | 0.79 | 0.12 | 0.19 | 0.02 | **0.57** | 0.25 |
| | ConceptFusion [11] | 0.74 | 0.10 | 0.26 | 0.02 | 0.63 | 0.30 |
| | OpenScene [23] | 0.77 | 0.06 | 0.18 | 0.01 | **0.57** | 0.31 |
| HM3D | ConceptGraphs [6] | 0.86 | 0.08 | 0.13 | 0.02 | 0.59 | 0.20 |
| | ConceptGraphs (GPT) [6] | 0.68 | **0.15** | 0.32 | **0.03** | 0.59 | 0.36 |
| | HOV-SG [35] | **0.88** | 0.12 | 0.11 | 0.02 | 0.59 | **0.19** |
| | Kassab2024 [12] | 0.80 | 0.06 | 0.19 | 0.01 | 0.62 | 0.26 |
| | OpenMask3D [32] | 0.86 | 0.10 | 0.12 | 0.02 | **0.56** | 0.20 |
| | ConceptFusion [11] | 0.78 | 0.07 | 0.20 | 0.02 | 0.61 | 0.27 |
| | OpenScene [23] | 0.87 | 0.05 | **0.10** | 0.01 | **0.56** | 0.22 |

Table 3: **Set Ranking Evaluation.** For $mR$, $R_S$, $R_{DVS}$ being the mean score and the respective inlier rates, higher is better. For the underscoring and overscoring penalties $P_S^{\leftarrow}$, $P_{DVS}^{\leftarrow}$, $P_{DVS}^{\rightarrow}$, lower is better.

| Data | Method | $mAP\uparrow$ | $AP_{50}\uparrow$ | $AP_{25}\uparrow$ |
|---|---|---|---|---|
| Replica | ConceptGraphs [6] | 5.86 | 11.32 | 22.39 |
| | ConceptGraphs (GPT) [6] | 5.13 | 10.77 | 18.19 |
| | HOV-SG [35] | 5.76 | 11.67 | **25.30** |
| | Kassab2024 [12] | 1.38 | 2.87 | 7.54 |
| | OpenMask3D + NMS [32] | **11.47** | **17.01** | 24.02 |
| ScanNet++ | ConceptGraphs [6] | 1.45 | 4.36 | 15.27 |
| | ConceptGraphs (GPT) [6] | 1.97 | 5.54 | 13.39 |
| | HOV-SG [35] | 1.79 | 4.95 | **18.75** |
| | Kassab2024 [12] | 0.40 | 1.19 | 3.39 |
| | OpenMask3D + NMS [32] | **4.00** | **6.90** | 10.34 |
| HM3D | ConceptGraphs [6] | **5.09** | **8.05** | **11.18** |
| | ConceptGraphs (GPT) [6] | 4.80 | 7.75 | 10.76 |
| | HOV-SG [35] | 3.44 | 5.39 | 7.42 |
| | Kassab2024 [12] | 1.03 | 1.87 | 3.97 |
| | OpenMask3D + NMS [32] | 4.03 | 5.56 | 8.35 |

Table 4: **Object Retrieval Evaluation.** All values are reported in percentage (%). NMS stands for Non-maximum Suppression and is used to select object masks in the OpenMask3D [32] pipeline.

## 4.2 Open-Set Object Retrieval

We report AP results in Tab. 4. Overall AP is low and in line with comparable benchmarks [5], highlighting the challenges of this task and the potential for improvement. OpenMask3D (with Non-Maximum Suppression and access to clean mesh inputs) leads the Replica and ScanNet++ metrics but fails to generalize to the larger HM3D scenes, where ConceptGraphs is the top-performing method. The performance of most methods significantly increases when queries include depiction information—a fact that can be directly observed in Supp. Fig. 14, where we provide separate metrics for *synonym* queries and *synonym-depiction* queries. The performance improvement is less noticeable in the case of ConceptGraphs (GPT), a possible sign that short image captions tend to prioritize synonyms. Those results emphasize the importance of a benchmark supporting queries for every object in a scene to prevent bias towards objects with prominent imagery.

For a more thorough analysis, we further report the queried object ranks in Supp. Fig. 15, with separate counts for queries with no predicted match at IoU .25. For a significant number of queries, methods fail to output predicted instances that significantly overlap with the ground truth. Properly segmenting instance geometry remains challenging for all considered methods.

## 5 Limitations

While our benchmark offers a comprehensive set of ground truth labels across diverse 3D scenes, it has certain limitations. Our label set does not account for additional object properties such as affordances, material, and color. It is restricted to the segments provided in the original RGB-D datasets, often omitting smaller parts of a larger object such as cupboard handles or sofa cushions. Our object retrieval query set inherits the same limitations and is generated using a simple template that could warrant further investigation.

## 6 Conclusion

We introduced OpenLex3D, a new benchmark for open-vocabulary evaluation that captures real-world language variability. Our benchmark includes human-annotated labels for three RGB-D datasets—ScanNet++, Replica, and HM3D—providing 13 times more labels per scene than the original annotations. We categorize our labels into multiple tiers of specificity, allowing for more nuanced evaluation. We assess the performance of both object-centric and dense representations on two key tasks: semantic segmentation and object retrieval. Our evaluation shows that no single method performs well across both tasks, indicating that there is scope for future improvement, particularly within feature fusion and segmentation strategies.

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
