## A  Acknowledgments

The authors would like to thank Ulrich-Michael, Frances, James, Maryam, and Mandolyn for their help in labeling the dataset. The work at the Université de Montréal was supported by the Natural Sciences and Engineering Research Council of Canada (NSERC) (Paull), an NSERC PGS D Scholarship (Morin) and an FRQNT Doctoral Scholarship (Morin). Moreover, this research was enabled in part by compute resources provided by Mila (mila.quebec). The work at the University of Freiburg was funded by an academic grant from NVIDIA. The work at the University of Oxford was supported by a Royal Society University Research Fellowship (Fallon, Kassab), a Sellafield Robotics and AI Centre of Excellence Grant, and EPSRC C2C Grant EP/Z531212/1 (Mattamala), and the National Research Foundation of Korea (NRF) grant funded by the Korea government (MSIT)(No. RS-2024-00461409).

## B  Limitations of Standard Semantic Segmentation Metrics

In this section, we elaborate on why standard semantic segmentation metrics such as mIoU become inadequate when dealing with larger label sets and open-vocabulary features. Although mIoU remains the prevailing metric in open-vocabulary semantic segmentation—where ground truth object labels are used as prompts—it struggles to remain informative as the label space expands. In Fig. 6, we compare the mIoU scores of five different methods using the original Replica ground truth labels (101 labels for all scenes) versus an extended set (759 labels for all scenes). The extended set of labels additionally contains all the depictions and visually similar OpenLex3D labels. As such, we injected plausible labels but not precise labels that capture failure modes of the actual prediction methods. In this way, we can study to which degree methods are subject to label perturbation and do not predict the actual intended label. The results show a marked drop in mIoU performance when these additional, plausible but imprecise labels are introduced. This reveals a key limitation: the methods are sensitive to the presence of such labels. A conventional segmentation evaluation with mIoU and a short prompt list fails to capture this nuance. As a result, it offers limited insights into feature quality and does not help in diagnosing specific failure modes.

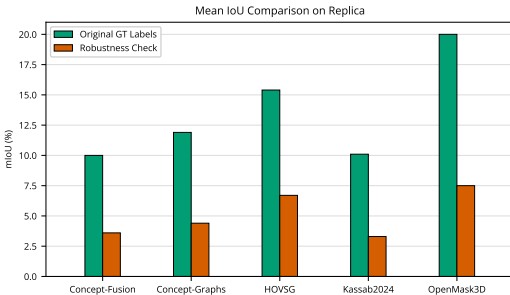

Figure 6: **Mean IoU comparison on five different methods on Replica.** We show the effect on mIoU when using the original ground truth prompt lists (101 labels) vs an extended prompt list (759 labels).

## C  Datasets - Further Details

In this appendix we provide details about the scenes selected from the parent datasets and their type (Tab. 5). Our goal was to include a diverse range of environments, covering various home and living spaces of different scales, as well as office and laboratory spaces.

## D  Labeling Instructions

We depict an example of the labeling interface employed when labeling a single object instance of the Replica dataset in Fig. 7.

The instructions provided to the human annotators were as follows:

| Dataset | Scene ID | Scene Type |
|---------|----------|------------|
| ScanNet++ | *0a76e06478* | Bedroom |
| | *0a7cc12c0e* | Studio apartment |
| | *1f7cbbdde1* | Studio apartment |
| | *49a82360aa* | Living area & office |
| | *4c5c60fa76* | Lab room |
| | *8a35ef3cfe* | Living area & kitchen |
| | *c0f5742640* | Kitchen |
| | *fd361ab85f* | Office |
| HM3D | *00824-Dd4bFSTQ8gi* | Single-story home |
| | *00829-QaLdnwvtxbs* | Single-story home |
| | *00843-DYehNKdT76V* | Two-story home |
| | *00847-bCPU9suPUw9* | Three-story home |
| | *00873-bxsVRursffK* | Two-story home |
| | *00877-4ok3usBNeis* | Two-story home |
| | *00890-6s7QHgap2fW* | Two-story home |
| Replica | *office0* | Meeting room |
| | *office1* | Common room |
| | *office2* | Meeting room |
| | *office3* | Meeting room |
| | *office4* | Meeting room |
| | *room0* | Living room |
| | *room1* | Bedroom |
| | *room2* | Dining room |

Table 5: **Full list of chosen scenes and scene types**. The scenes are from a variety of environments including studio apartments and multi-storey homes, offices, meeting rooms, and labs. All scenes are available for non-commercial research purposes. See the Replica Research Terms, the HM3D Terms and Conditions and the ScanNet++ Terms of Use.

*You will be presented with multiple views of an object. We removed the background in the top left view to highlight the relevant object. Please label the object according to the following three category system:*

**Synonyms.** *Assign the most accurate label for the object, along with any closely related synonyms. For example:*
- `sofa, couch, settee`
- `screen, monitor, computer`
- `glasses, spectacles`
*For bottles and containers, consider adding both the content and a reference to the container. For example:* `ketchup, ketchup bottle`

**Depictions.** *If the object features a distinct pattern, image, or design of another object or concept, label it accordingly. For example, if you see a painting, figurine or toy representing a cat or a book on cats, you might use:*
- `cat`
*You can also add to this category legible words or brands.*

**Visually Similar Objects.** *Identify objects that are not synonyms but appear visually similar to the highlighted object. Consider objects of similar color or shape. Visually similar labels should be wrong, but have an understandable visual connection with the displayed object.*

*Examples corresponding to the objects mentioned in Synonyms might include:*
- `chair, armchair`
- `television`
- `sunglasses, goggles`

**Notes:**
*- Please separate each input with a comma (,).*
*- You can use multiple words as a label. Please separate words belonging to the same label with a space ( ). For example "coffee mug" or "ice cream".*

*- You are encouraged to input as many appropriate labels as you can think of for each level.*
*- If you cannot think of any appropriate labels for any level, simply leave that level blank.*
*- You can leave all levels blank.*

***Additional Notes for ScanNet++ Scenes:***
*- Some crops may show a magenta mask to protect privacy. Please ignore it.*
*- If a crop covers multiple objects, please list them separately. If there are too many objects, focus on the main ones.*
*- If the relevant object(s) is unclear because the crops are too blurry or showing different objects, please leave all the fields blank.*

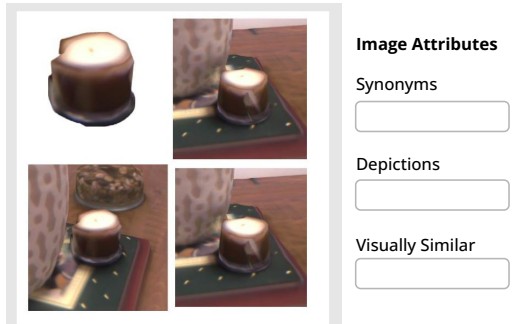

Figure 7: **The segments.ai web interface showing an example provided image.** The top left image is segmented using SAM [14] and aims to highlight the intended object for labeling more clearly to the annotator. The other images provide additional context.

# E  Label Distributions

Tab. 1 provides a breakdown of the number and variety of labels present in each dataset as well as the size of the prompt lists used for the semantic segmentation task and the number of queries in the object retrieval task.

The labels captured in the dataset encompass a broad range of semantically similar terms, such as "pillow" and "cushion", or "sofa" and "couch". By incorporating this large range of synonyms, our benchmark aligns well with real-world variability in object naming. This allows for a deeper level analysis and evaluation of open-vocabulary representations. We show the distribution of the *synonyms* category for Replica in Fig. 9, ScanNet++ in Fig. 10 and HM3D in Fig. 11. We also show a visual example of a subset of words connected to the label *desk* in Fig. 8. The edges are added between words that appear as labels for the same object regardless of category. We also show additional edges added between words that appear often within the same category.

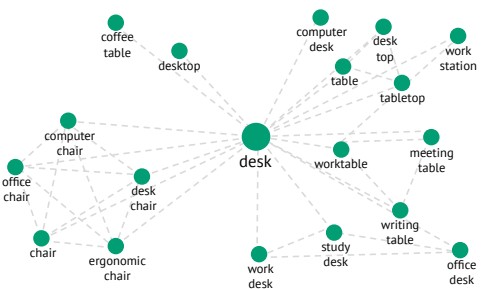

Figure 8: **A subset of the labels connected to the word *desk* in the OpenLex3D synonym label set.** Our labels encompass the diversity of real-world object names, allowing more realistic evaluation of open-vocabulary representations.

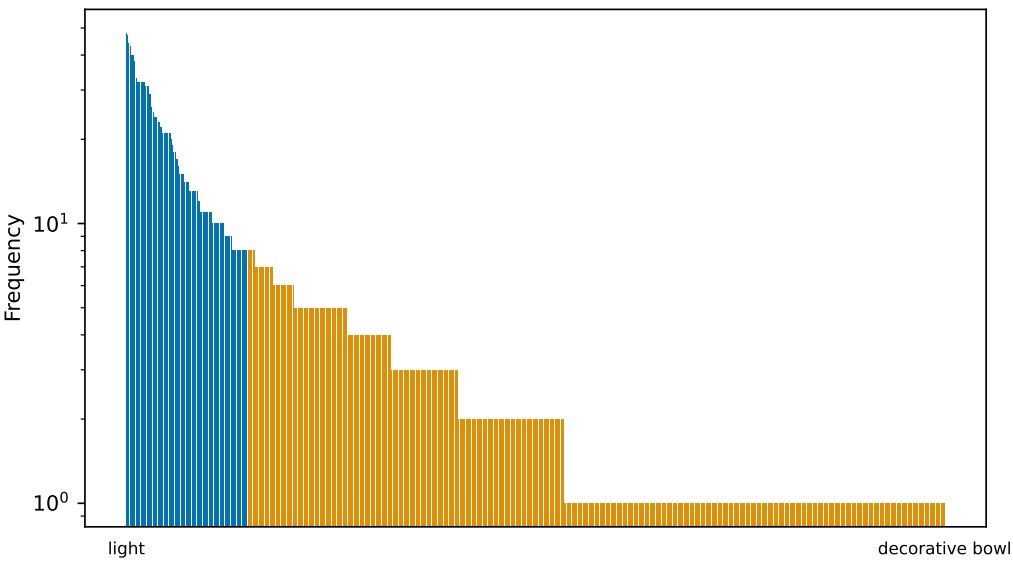

(a) OpenLex3D Synonym Distribution for the Replica Dataset

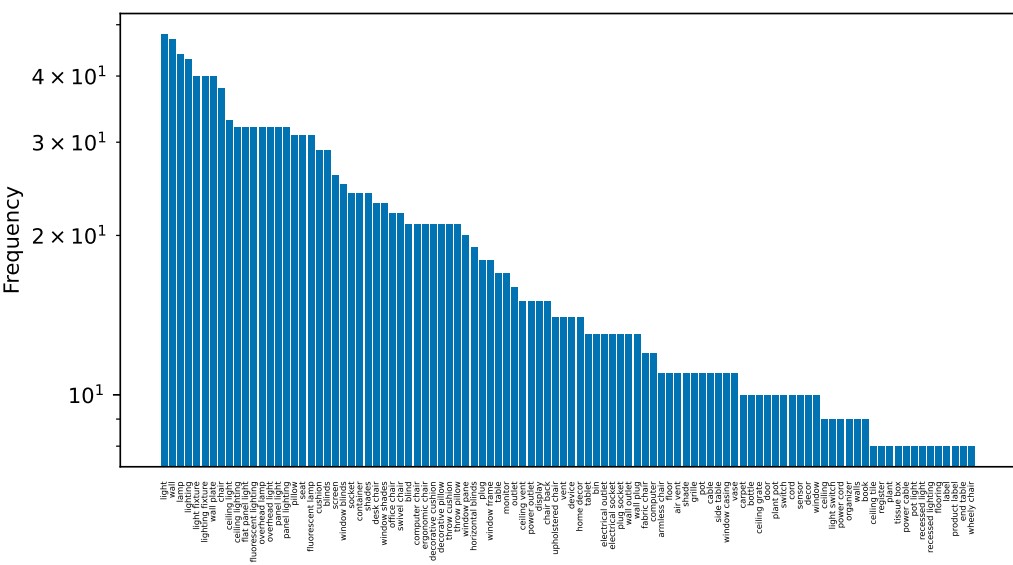

(b) OpenLex3D Top 100 Synonym Distribution for the Replica Dataset

Figure 9: **Replica Label Distribution**. We show the label distribution for (a) all labels in the *synonyms* category (a total of 673 unique synonyms across all scenes) and (b) the top 100 labels in the *synonyms* category. The y axis is plotted as a log scale and only the head and tail labels are shown for the full set of synonyms for readability.

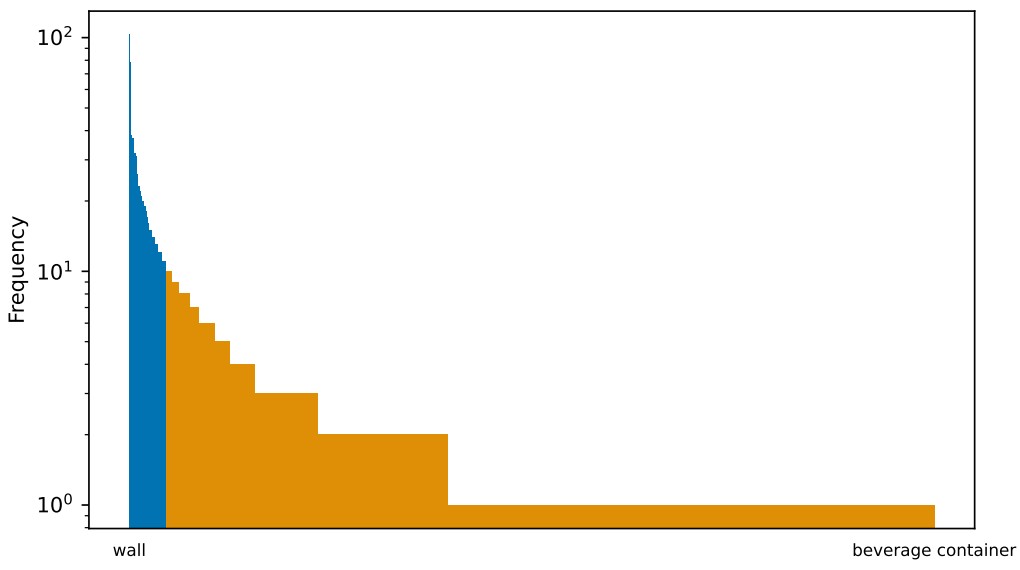

(a) OpenLex3D Synonym Distribution for the ScanNet++ Dataset

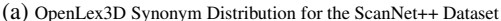

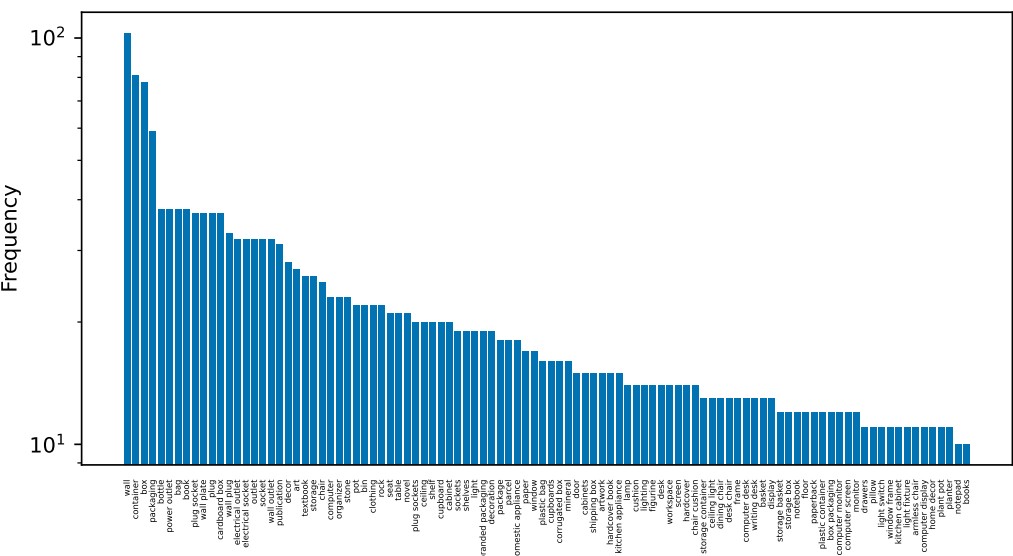

(b) OpenLex3D Top 100 Synonym Distribution for the ScanNet++ Dataset

Figure 10: **ScanNet++ Label Distribution**. We show the label distribution for (a) all labels in the *synonyms* category (a total of 2154 unique synonyms across all scenes) and (b) the top 100 labels in the *synonyms* category. The y axis is plotted as a log scale and only the head and tail labels are shown for the full set of synonyms for readability.

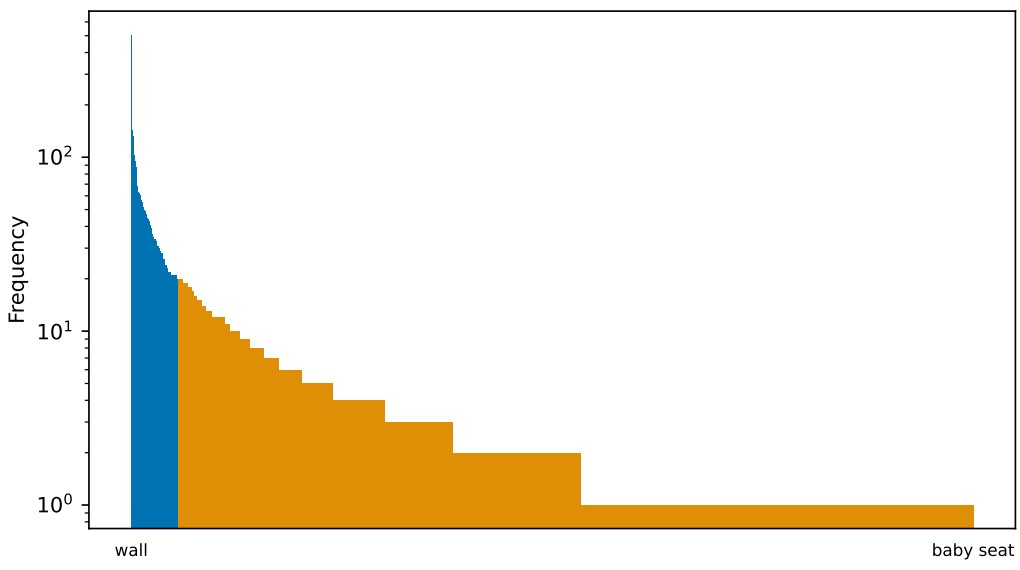

(a) OpenLex3D Synonym Distribution for the HM3D Dataset

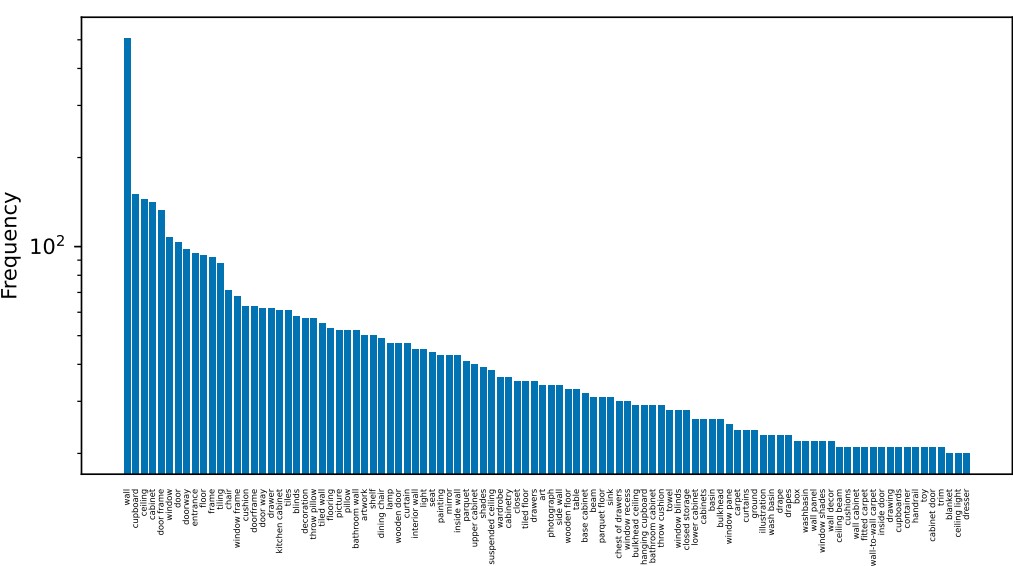

(b) OpenLex3D Top 100 Synonym Distribution for the HM3D Dataset

Figure 11: **HM3D Label Distribution**. We show the label distribution for (a) all labels in the *synonyms* category (a total of 1789 unique synonyms across all scenes) and (b) the top 100 labels in the *synonyms* category. The y axis is plotted as a log scale and only the head and tail labels are shown for the full set of synonyms for readability.

# F  Additional Details on Set Ranking Metric

In the following, we formalize the left and right box constraint penalties which were mentioned in Sec. 3.2.1. As the *synonyms* are considered the primary target-rank set, we only need to consider a violation of its right box constraint due to potential underscoring of *synonyms*, which is formalized as:

$$R_S^{\text{right}} = 1 - \frac{1}{|\mathcal{P}_*|} \sum_{p \in \mathcal{P}_*} \left( \frac{1}{|\mathcal{Y}_S^p|} \sum_i^{|\mathcal{Y}_S^p|} \left( 1 + \min\left(0, \frac{r_i - b_l^S}{b_l^S}\right) \right) \right), \tag{5}$$

where $\mathcal{Y}_S^p$ represents the set of synonym labels of each object $p \in \mathcal{P}_*$. Similarly, we compute the left and right box-constraint scores for the $DVS$ category set as

$$R_{DVS}^{\text{left}} = \frac{1}{|\mathcal{P}_*|} \sum_{p \in \mathcal{P}_*} \left( \frac{1}{|\mathcal{Y}_{DVS}^p|} \sum_i^{|\mathcal{Y}_{DVS}^p|} \left( 1 - \max\left(0, \frac{r_i - b_r^{DVS}}{|\mathcal{L}^p| - b_r^{DVS}}\right) \right) \right), \tag{6}$$

$$R_{DVS}^{\text{right}} = \frac{1}{|\mathcal{P}_*|} \sum_{p \in \mathcal{P}_*} \left( \frac{1}{|\mathcal{Y}_{DVS}^p|} \sum_i^{|\mathcal{Y}_{DVS}^p|} \left( 1 + \min\left(0, \frac{r_i - b_l^{DVS}}{b_l^{DVS}}\right) \right) \right) \tag{7}$$

$$\tag{8}$$

where $\mathcal{Y}_{DVS}^p$ represents the set of $DVS$ labels of each object $p \in \mathcal{P}_*$. Since those scores measure normalized rank similarity rather than quantifying a penalty, we inverse them as follows:

$$P_S^{\leftarrow} = 1 - R_S^{\text{right}}, \tag{9}$$

$$P_{DVS}^{\leftarrow} = 1 - R_{DVS}^{\text{left}}, \tag{10}$$

$$P_{DVS}^{\rightarrow} = 1 - R_{DVS}^{\text{right}} \tag{11}$$

Thus, $P_S^{\leftarrow}$ quantifies the *synonym* underscoring penalty, $P_{DVS}^{\leftarrow}$ quantifies the $DVS$ overscoring penalty, whereas $P_{DVS}^{\rightarrow}$ quantifies the $DVS$ underscoring penalty.

# G  Implementation Details

**Kassab2024.**   We follow the execution of the pipeline described in Kassab et al. [12]. We chose intervals of 25 frames, using region-growing parameters of 100 considered neighbors, a smoothness threshold of 0.05 radians, a curvature threshold of 1, and a tree-based search method. The entropy was calculated using the 100-label Replica prompt list, as described in the paper.

**ConceptGraphs.**   We ran ConceptGraphs [6] with a stride of 10 frames and at a similarity threshold of 0.75.

**ConceptGraphs (GPT).**   As in the original ConceptGraphs paper, we rely on a VLM to extract object captions from multiple views. We streamline captioning by providing `gpt-4o` with the 4 best object views (based on SAM segment area) and prompt it to give a succinct description of the central object:

> *You are a helpful assistant that describes objects in a single word. You will be provided with up to 4 views of the same object. Describe the object with a few words. You should generally use a single word, but you can use more than one if needed (e.g., ice cream, toilet paper).*

We embed the resulting description with CLIP and use this instead of the standard ConceptGraphs features. The representation is otherwise identical to ConceptGraphs.

**HOV-SG.**   We run the publicly-available HOV-SG [35] pipeline using a stride of 10 frames and its sequential merge paradigm. All other parameters remain unchanged.

**OpenMask3D.** OpenMask3D [32] was executed at intervals of 10 frames. For the open-set semantic segmentation task, points belonging to multiple masks, the mask with the highest score was chosen for that point. For the open-set object retrieval task, non-maximum suppression (with an IoU threshold of 0.50) was applied to all masks. All the resultant masks were then used for AP calculation.

**ConceptFusion.** We ran ConceptFusion [11] with a default stride of 10 frames, increasing it as needed for larger scenes to fit in memory. We used the SAM version and default parameters from the repository.

**OpenScene.** OpenScene [23] was executed at intervals of 10 frames. For evaluation, we use the multi-view 2D fused features, as they better align with the other methods.

# H  Experiments

## H.1  Top N Freq. - Further Results

We present Top 1 and Top 10 quantitative frequency results in Tab. 6. We also show qualitative comparisons in Fig. 12 for OpenMask3D and ConceptFusion.

As N increases $FREQ^S$ increases, and $FREQ^I$ decreases with all other categories experiencing minor changes of approximately 0.01. Relaxing the metric to N = 5 helps mitigate against minor inconsistencies in the ground truth data. For example, in Fig. 12, OpenMask3D predicts "smart board screen" for the top 1 prediction which is categorized as *clutter*. This occurs because a nearby screen is labeled "smart board screen", while the target object is not. By considering the top 5 predictions, the correct label, "smart board" is found, re-categorizing the prediction as a *synonym*.

By increasing N we account for ambiguities in language descriptions, while also ensuring that entirely incorrect predictions remain unchanged, as demonstrated for ConceptFusion in Fig. 12 (bottom). We select N = 5 as an optimal balance between accomodating label ambiguity and preventing the metric from being overly permissive, which could lead to incorrect predictions being classified as *synonyms* or other categories. Additional qualitative results at N = 5 are shown in Figure 13.

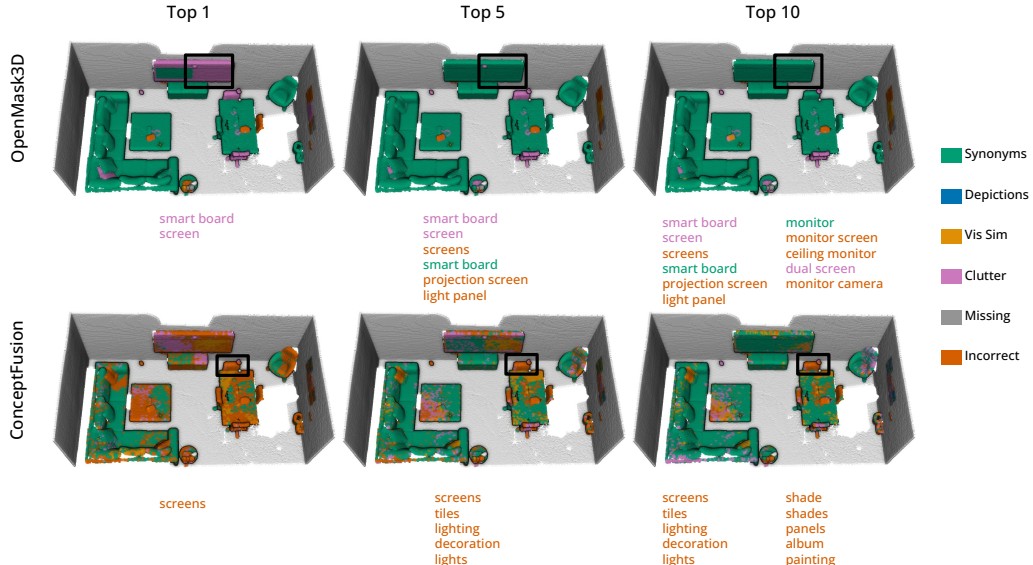

Figure 12: **Top 1, 5 and 10 Comparison for OpenMask3D [32] and ConceptFusion [11] on Replica [31] Office 2.** Increasing N allows for ambiguities in semantically similar labels (top) while also ensuring that incorrectly predicted labels remain incorrect (bottom).

| N | Dataset | Method | S↑ | D↓ | VS↓ | C↓ | M↓ | I↓ |
|---|---------|--------|----|----|-----|----|----|----|
| 1 | Replica | ConceptGraphs | 0.18 | 0.01 | 0.06 | **0.13** | **0.02** | 0.61 |
| | | ConceptGraphs (GPT) | **0.37** | 0.03 | 0.03 | 0.14 | **0.02** | 0.41 |
| | | HOV-SG | 0.25 | 0.01 | 0.03 | 0.22 | 0.07 | 0.42 |
| | | Kassab2024 | 0.13 | **0.00** | 0.04 | 0.28 | 0.11 | 0.45 |
| | | OpenMask3D | 0.27 | **0.00** | **0.02** | 0.26 | 0.10 | **0.35** |
| | | ConceptFusion | 0.18 | 0.01 | 0.06 | **0.13** | **0.02** | 0.61 |
| | | OpenScene | 0.20 | **0.00** | 0.04 | 0.20 | 0.07 | 0.49 |
| | ScanNet++ | ConceptGraphs | 0.12 | 0.01 | 0.02 | 0.05 | 0.13 | 0.68 |
| | | ConceptGraphs (GPT) | **0.33** | 0.01 | 0.03 | 0.08 | 0.13 | **0.42** |
| | | HOV-SG | 0.2 | 0.02 | 0.03 | 0.09 | 0.08 | 0.59 |
| | | Kassab2024 | 0.05 | **0.00** | **0.01** | 0.08 | 0.38 | 0.49 |
| | | OpenMask3D | 0.15 | 0.01 | **0.01** | 0.17 | 0.13 | 0.53 |
| | | ConceptFusion | 0.15 | 0.01 | 0.02 | **0.04** | **0.04** | 0.74 |
| | | OpenScene | 0.07 | **0.00** | **0.01** | 0.11 | 0.22 | 0.59 |
| | HM3D | ConceptGraphs | 0.10 | 0.01 | **0.01** | **0.04** | 0.08 | 0.75 |
| | | ConceptGraphs (GPT) | 0.32 | **0.00** | 0.03 | 0.09 | 0.09 | **0.47** |
| | | HOV-SG | 0.13 | 0.01 | 0.02 | 0.07 | 0.08 | 0.69 |
| | | Kassab2024 | 0.08 | **0.00** | 0.01 | 0.05 | 0.22 | 0.65 |
| | | OpenMask3D | **0.15** | **0.00** | 0.01 | 0.07 | 0.29 | 0.48 |
| | | ConceptFusion | 0.09 | 0.01 | 0.01 | **0.03** | 0.08 | 0.79 |
| | | OpenScene | 0.07 | **0.00** | **0.00** | 0.04 | **0.06** | 0.83 |
| 10 | Replica | ConceptGraphs | 0.52 | 0.01 | 0.12 | 0.23 | 0.02 | 0.11 |
| | | ConceptGraphs (GPT) | 0.52 | 0.02 | **0.04** | 0.23 | 0.02 | 0.17 |
| | | HOV-SG | 0.51 | **0.00** | 0.07 | 0.27 | 0.07 | 0.08 |
| | | Kassab2024 | 0.33 | **0.00** | 0.08 | 0.24 | 0.12 | 0.22 |
| | | OpenMask3D | 0.51 | 0.01 | 0.07 | 0.26 | 0.10 | 0.06 |
| | | ConceptFusion | 0.40 | 0.01 | 0.10 | **0.20** | **0.00** | 0.29 |
| | | OpenScene | **0.55** | **0.00** | 0.05 | 0.29 | 0.07 | **0.04** |
| | ScanNet++ | ConceptGraphs | 0.35 | 0.02 | 0.06 | 0.12 | 0.13 | 0.32 |
| | | ConceptGraphs (GPT) | **0.48** | 0.01 | 0.04 | 0.11 | 0.13 | 0.23 |
| | | HOV-SG | **0.48** | 0.01 | 0.05 | 0.16 | 0.08 | **0.22** |
| | | Kassab2024 | 0.15 | **0.00** | **0.03** | 0.19 | 0.38 | 0.25 |
| | | OpenMask3D | 0.35 | 0.01 | 0.04 | 0.31 | 0.13 | 0.17 |
| | | ConceptFusion | 0.35 | 0.01 | 0.04 | **0.09** | **0.04** | 0.46 |
| | | OpenScene | 0.21 | **0.00** | **0.03** | 0.29 | 0.22 | 0.25 |
| | HM3D | ConceptGraphs | 0.34 | 0.02 | 0.04 | 0.16 | **0.08** | 0.35 |
| | | ConceptGraphs (GPT) | **0.53** | **0.00** | 0.03 | 0.12 | 0.09 | 0.23 |
| | | HOV-SG | 0.43 | 0.02 | 0.09 | 0.20 | **0.08** | 0.23 |
| | | Kassab2024 | 0.25 | 0.01 | **0.02** | 0.17 | 0.23 | 0.33 |
| | | OpenMask3D | 0.42 | **0.00** | 0.02 | 0.13 | 0.29 | **0.13** |
| | | ConceptFusion | 0.29 | 0.01 | 0.03 | **0.10** | **0.08** | 0.48 |
| | | OpenScene | 0.27 | **0.00** | 0.02 | 0.25 | 0.06 | 0.39 |

Table 6: **Freq. Top 1 and 10 Results for Object-Centric and Dense Representations.** Where *S* is synonyms, *D* is depictions, *VS* is visually similar, *C* is clutter, *M* is missing and *I* is incorrect.

## H.2  Object Retrieval - Further Results

We provide additional context for the object retrieval experiment in Fig. 14 and Fig. 15.

## H.3  Compute Resources

We ran our final experiments on an academic compute cluster for efficiency. However, we expect our benchmark and most of the considered methods to run on consumer-grade hardware. We developed a good part of the OpenLex3D codebase and computed some early benchmark results on a workstation with a 16-Core AMD Ryzen CPU, an NVIDIA GeForce RTX 2080 Ti and 32 GB of RAM.

## H.4  Standard Deviations for All Metrics

We include standard deviations for all metrics in Tables 7, 8 and 9. The standard deviations are non-negligible. For example, there is high variance in the clutter score for OpenMask3D. OpenMask3D relies on a fixed-class 3D segmentation model, Mask3D, and is therefore influenced by the objects which appear in the original training sets and may not generalize to out-of -distribution scenes.

| N | Dataset | Method | $S\uparrow$ | $D\downarrow$ | $VS\downarrow$ | $C\downarrow$ | $M\downarrow$ | $I\downarrow$ |
|---|---------|--------|------|------|------|------|------|------|
| | | ConceptGraphs | 0.10 | 0.01 | 0.05 | 0.07 | 0.01 | 0.12 |
| | | ConceptGraphs (GPT) | 0.11 | 0.02 | 0.03 | 0.10 | 0.01 | 0.15 |
| | | HOV-SG | 0.08 | 0.01 | 0.02 | 0.13 | 0.04 | 0.16 |
| | Replica | Kassab2024 | 0.06 | 0.00 | 0.06 | 0.11 | 0.03 | 0.13 |
| | | OpenMask3D | 0.10 | 0.01 | 0.04 | 0.13 | 0.07 | 0.15 |
| | | ConceptFusion | 0.06 | 0.00 | 0.04 | 0.03 | 0.01 | 0.08 |
| | | OpenScene | 0.08 | 0.00 | 0.04 | 0.06 | 0.03 | 0.10 |
| | | ConceptGraphs | 0.03 | 0.01 | 0.01 | 0.03 | 0.05 | 0.04 |
| | | ConceptGraphs (GPT) | 0.07 | 0.01 | 0.02 | 0.04 | 0.05 | 0.08 |
| | | HOV-SG | 0.05 | 0.01 | 0.01 | 0.04 | 0.03 | 0.06 |
| 1 | ScanNet++ | Kassab2024 | 0.03 | 0.00 | 0.01 | 0.05 | 0.08 | 0.06 |
| | | OpenMask3D | 0.08 | 0.01 | 0.01 | 0.08 | 0.05 | 0.09 |
| | | ConceptFusion | 0.05 | 0.01 | 0.01 | 0.02 | 0.02 | 0.06 |
| | | OpenScene | 0.06 | 0.00 | 0.01 | 0.11 | 0.31 | 0.23 |
| | | ConceptGraphs | 0.03 | 0.01 | 0.01 | 0.01 | 0.02 | 0.03 |
| | | ConceptGraphs (GPT) | 0.05 | 0.01 | 0.01 | 0.04 | 0.02 | 0.06 |
| | | HOV-SG | 0.03 | 0.01 | 0.01 | 0.03 | 0.03 | 0.03 |
| | HM3D | Kassab2024 | 0.02 | 0.00 | 0.01 | 0.02 | 0.02 | 0.02 |
| | | OpenMask3D | 0.03 | 0.00 | 0.01 | 0.04 | 0.03 | 0.07 |
| | | ConceptFusion | 0.02 | 0.01 | 0.01 | 0.01 | 0.05 | 0.07 |
| | | OpenScene | 0.03 | 0.00 | 0.00 | 0.02 | 0.01 | 0.04 |
| | | ConceptGraphs | 0.14 | 0.02 | 0.07 | 0.08 | 0.01 | 0.11 |
| | | ConceptGraphs (GPT) | 0.13 | 0.02 | 0.02 | 0.10 | 0.01 | 0.15 |
| | | HOV-SG | 0.05 | 0.00 | 0.03 | 0.11 | 0.04 | 0.09 |
| | Replica | Kassab2024 | 0.07 | 0.00 | 0.04 | 0.08 | 0.03 | 0.11 |
| | | OpenMask3D | 0.08 | 0.02 | 0.10 | 0.11 | 0.07 | 0.05 |
| | | ConceptFusion | 0.07 | 0.01 | 0.05 | 0.03 | 0.01 | 0.09 |
| | | OpenScene | 0.11 | 0.00 | 0.04 | 0.11 | 0.03 | 0.10 |
| | | ConceptGraphs | 0.04 | 0.02 | 0.03 | 0.05 | 0.05 | 0.07 |
| | | ConceptGraphs (GPT) | 0.05 | 0.01 | 0.02 | 0.05 | 0.05 | 0.05 |
| | | HOV-SG | 0.05 | 0.01 | 0.02 | 0.07 | 0.03 | 0.07 |
| 5 | ScanNet++ | Kassab2024 | 0.04 | 0.00 | 0.02 | 0.08 | 0.08 | 0.09 |
| | | OpenMask3D | 0.10 | 0.01 | 0.02 | 0.13 | 0.05 | 0.11 |
| | | ConceptFusion | 0.07 | 0.01 | 0.01 | 0.03 | 0.02 | 0.08 |
| | | OpenScene | 0.10 | 0.00 | 0.01 | 0.16 | 0.31 | 0.16 |
| | | ConceptGraphs | 0.07 | 0.02 | 0.02 | 0.03 | 0.02 | 0.07 |
| | | ConceptGraphs (GPT) | 0.07 | 0.01 | 0.01 | 0.03 | 0.02 | 0.08 |
| | | HOV-SG | 0.09 | 0.01 | 0.02 | 0.03 | 0.03 | 0.08 |
| | HM3D | Kassab2024 | 0.02 | 0.01 | 0.00 | 0.04 | 0.02 | 0.02 |
| | | OpenMask3D | 0.05 | 0.01 | 0.01 | 0.03 | 0.03 | 0.04 |
| | | ConceptFusion | 0.02 | 0.01 | 0.01 | 0.01 | 0.05 | 0.07 |
| | | OpenScene | 0.07 | 0.00 | 0.01 | 0.05 | 0.01 | 0.09 |
| | | ConceptGraphs | 0.14 | 0.01 | 0.08 | 0.09 | 0.01 | 0.07 |
| | | ConceptGraphs (GPT) | 0.13 | 0.02 | 0.02 | 0.06 | 0.01 | 0.11 |
| | | HOV-SG | 0.05 | 0.00 | 0.04 | 0.08 | 0.04 | 0.05 |
| | Replica | Kassab2024 | 0.09 | 0.00 | 0.05 | 0.10 | 0.03 | 0.11 |
| | | OpenMask3D | 0.09 | 0.01 | 0.06 | 0.09 | 0.07 | 0.06 |
| | | ConceptFusion | 0.07 | 0.01 | 0.05 | 0.05 | 0.01 | 0.06 |
| | | OpenScene | 0.10 | 0.00 | 0.02 | 0.10 | 0.03 | 0.03 |
| | | ConceptGraphs | 0.04 | 0.01 | 0.03 | 0.05 | 0.05 | 0.07 |
| | | ConceptGraphs (GPT) | 0.03 | 0.01 | 0.02 | 0.06 | 0.05 | 0.06 |
| | | HOV-SG | 0.04 | 0.02 | 0.02 | 0.07 | 0.03 | 0.07 |
| 10 | ScanNet++ | Kassab2024 | 0.05 | 0.00 | 0.02 | 0.09 | 0.08 | 0.07 |
| | | OpenMask3D | 0.12 | 0.01 | 0.03 | 0.16 | 0.05 | 0.11 |
| | | ConceptFusion | 0.07 | 0.02 | 0.02 | 0.03 | 0.02 | 0.07 |
| | | OpenScene | 0.12 | 0.00 | 0.02 | 0.19 | 0.31 | 0.12 |
| | | ConceptGraphs | 0.08 | 0.02 | 0.02 | 0.03 | 0.02 | 0.06 |
| | | ConceptGraphs (GPT) | 0.07 | 0.01 | 0.02 | 0.02 | 0.02 | 0.07 |
| | | HOV-SG | 0.10 | 0.01 | 0.12 | 0.03 | 0.03 | 0.06 |
| | HM3D | Kassab2024 | 0.04 | 0.01 | 0.01 | 0.03 | 0.02 | 0.01 |
| | | OpenMask3D | 0.06 | 0.00 | 0.01 | 0.03 | 0.03 | 0.04 |
| | | ConceptFusion | 0.04 | 0.01 | 0.02 | 0.02 | 0.05 | 0.06 |
| | | OpenScene | 0.08 | 0.00 | 0.01 | 0.07 | 0.01 | 0.08 |

Table 7: **Standard Deviation of Freq. Results** Where $S$ is synonyms, $D$ is depictions, $VS$ is visually similar, $C$ is clutter, $M$ is missing and $I$ is incorrect.

| Data | Method | $mR\uparrow$ | $R_S\uparrow$ | $P_S^{\leftarrow}\downarrow$ | $R_{DVS}\uparrow$ | $P_{DVS}^{\leftarrow}\downarrow$ | $P_{DVS}^{\rightarrow}\downarrow$ |
|---|---|---|---|---|---|---|---|
| | ConceptGraphs | 0.03 | 0.06 | 0.04 | 0.02 | 0.06 | 0.04 |
| | ConceptGraphs (GPT) | 0.05 | 0.07 | 0.05 | 0.02 | 0.06 | 0.05 |
| | HOV-SG | 0.03 | 0.04 | 0.03 | 0.02 | 0.13 | 0.03 |
| Replica | Kassab2024 | 0.04 | 0.04 | 0.05 | 0.02 | 0.12 | 0.05 |
| | OpenMask3D | 0.02 | 0.05 | 0.03 | 0.02 | 0.07 | 0.01 |
| | ConceptFusion | 0.04 | 0.03 | 0.04 | 0.01 | 0.10 | 0.04 |
| | OpenScene | 0.03 | 0.05 | 0.03 | 0.02 | 0.12 | 0.04 |
| | ConceptGraphs | 0.02 | 0.02 | 0.03 | 0.01 | 0.11 | 0.02 |
| | ConceptGraphs (GPT) | 0.03 | 0.01 | 0.05 | 0.01 | 0.08 | 0.03 |
| | HOV-SG | 0.02 | 0.03 | 0.03 | 0.01 | 0.08 | 0.01 |
| ScanNet++ | Kassab2024 | 0.02 | 0.01 | 0.03 | 0.00 | 0.10 | 0.02 |
| | OpenMask3D | 0.03 | 0.06 | 0.04 | 0.01 | 0.11 | 0.03 |
| | ConceptFusion | 0.03 | 0.03 | 0.03 | 0.01 | 0.09 | 0.03 |
| | OpenScene | 0.03 | 0.03 | 0.05 | 0.01 | 0.11 | 0.04 |
| | ConceptGraphs | 0.02 | 0.02 | 0.02 | 0.00 | 0.07 | 0.02 |
| | ConceptGraphs (GPT) | 0.03 | 0.02 | 0.04 | 0.01 | 0.06 | 0.05 |
| | HOV-SG | 0.03 | 0.02 | 0.03 | 0.01 | 0.11 | 0.03 |
| HM3D | Kassab2024 | 0.02 | 0.01 | 0.01 | 0.01 | 0.14 | 0.03 |
| | OpenMask3D | 0.02 | 0.02 | 0.01 | 0.01 | 0.08 | 0.03 |
| | ConceptFusion | 0.02 | 0.01 | 0.02 | 0.00 | 0.16 | 0.02 |
| | OpenScene | 0.03 | 0.02 | 0.02 | 0.00 | 0.08 | 0.03 |

Table 8: **Set Ranking Standard Deviation Results.**

| Data | Method | $mAP\uparrow$ | $AP_{50}\uparrow$ | $AP_{25}\uparrow$ |
|---|---|---|---|---|
| | ConceptGraphs | 1.60 | 3.21 | 3.64 |
| | ConceptGraphs (GPT) | 3.64 | 5.51 | 6.58 |
| Replica | HOV-SG | 3.27 | 2.87 | 5.84 |
| | Kassab2024 | 1.60 | 3.21 | 3.64 |
| | OpenMask3D + NMS | 5.06 | 6.02 | 6.33 |
| | ConceptGraphs | 0.81 | 2.27 | 4.20 |
| | ConceptGraphs (GPT) | 1.85 | 3.02 | 3.89 |
| ScanNet++ | HOV-SG | 1.48 | 2.98 | 6.17 |
| | Kassab2024 | 0.33 | 1.26 | 2.01 |
| | OpenMask3D + NMS | 3.77 | 6.48 | 8.17 |
| | ConceptGraphs | 2.33 | 3.06 | 3.58 |
| | ConceptGraphs (GPT) | 1.00 | 1.60 | 1.59 |
| HM3D | HOV-SG | 1.13 | 1.87 | 2.35 |
| | Kassab2024 | 0.82 | 1.16 | 1.93 |
| | OpenMask3D + NMS | 1.77 | 2.38 | 2.88 |

Table 9: **Standard Deviation of Object Retreival Results**. NMS stands for Non-maximum Suppression and is used to select object masks in the OpenMask3D pipeline.

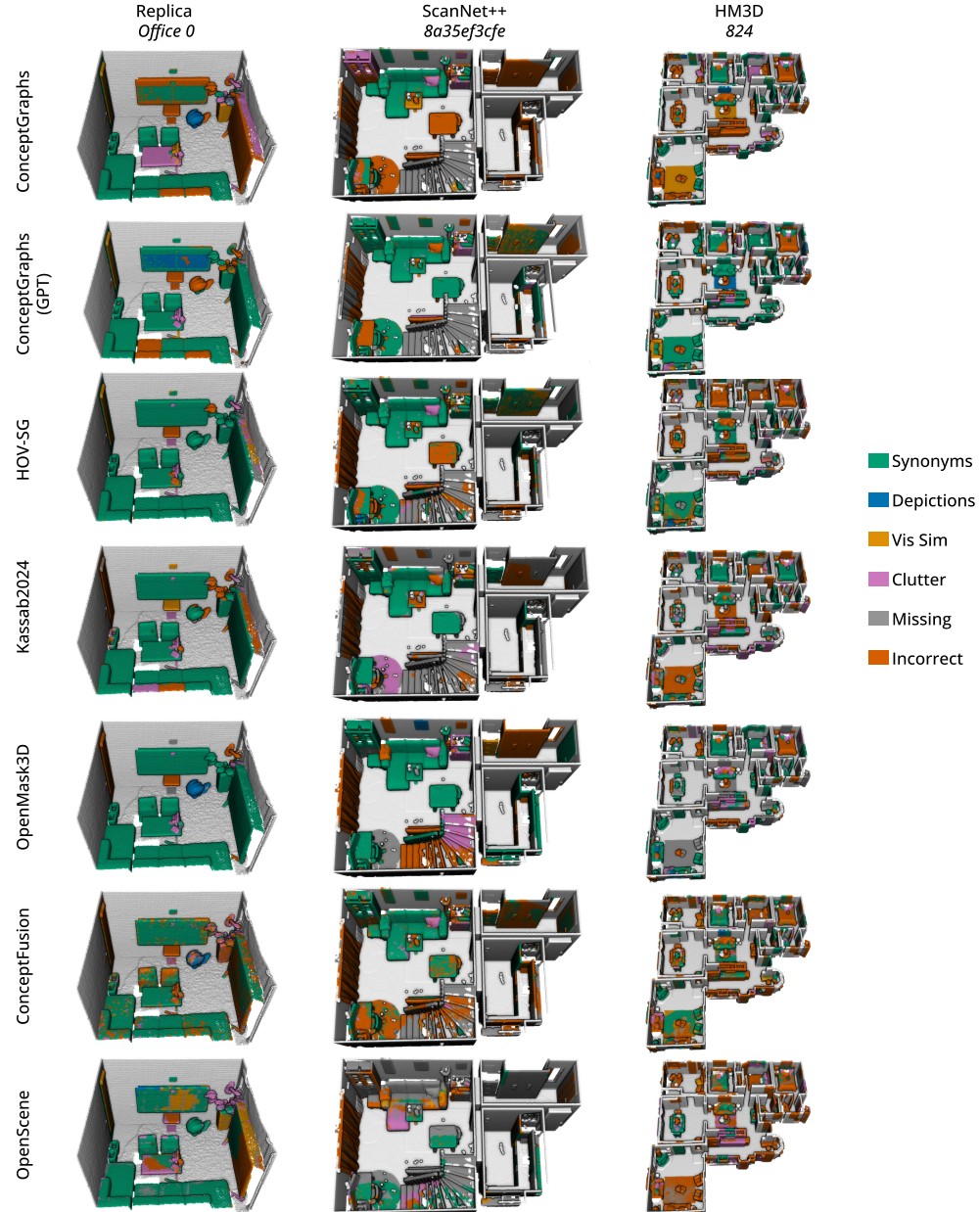

Figure 13: **Qualitative Top 5 Freq. Results.** We show results from all assessed methods on a selected scene from each dataset. The scenes range from small-scale reconstructed environments (Replica [31]) to large-scale cluttered home environments (HM3D [26]).

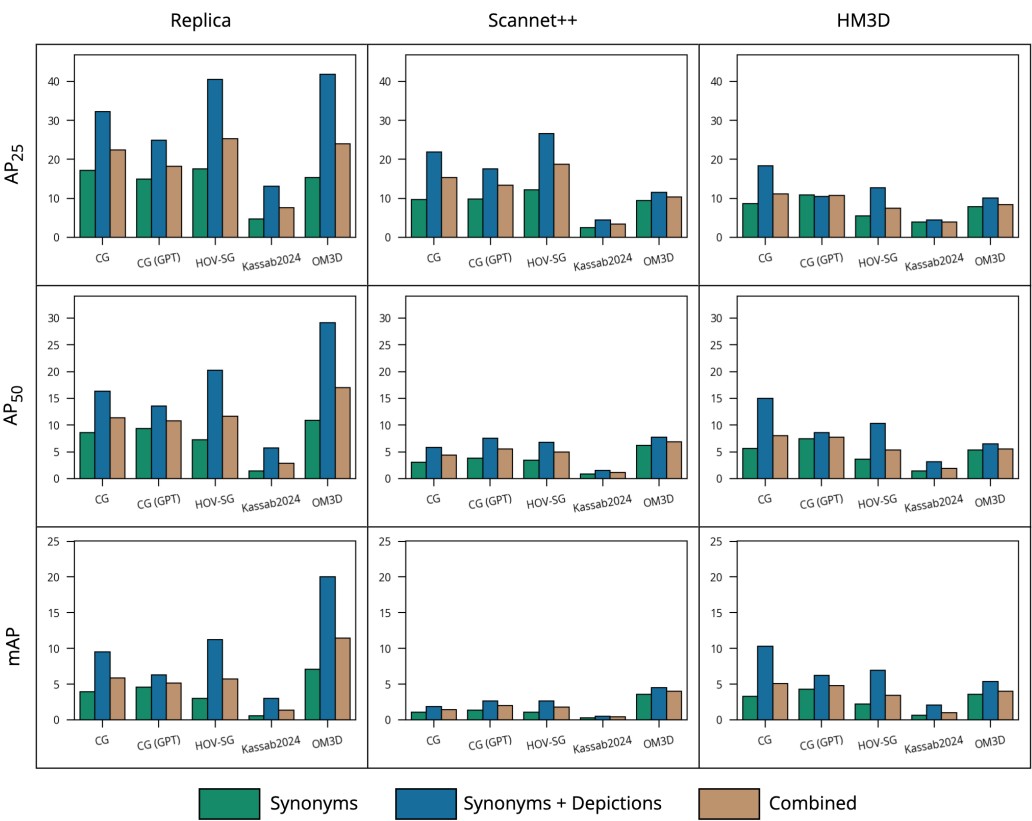

Figure 14: **AP by Category.** We break down the query metrics to distinguish *synonym*-only queries and queries formed from both a *synonym* and a *depiction*. CG, CG (GPT), and OM3D respectively refer to ConceptGraphs [6], ConceptGraphs (GPT) [6], and OpenMask3D [32]. Depictions tend to be more specific and feature salient concepts that are likely unique in the scene (e.g., a fisherman on a pillow in a Replica [31] room). We posit that this explains the significant gap in performance in favor of depiction-based queries. This experiment underscores the importance of a benchmark that provides queries for every object in a scene.

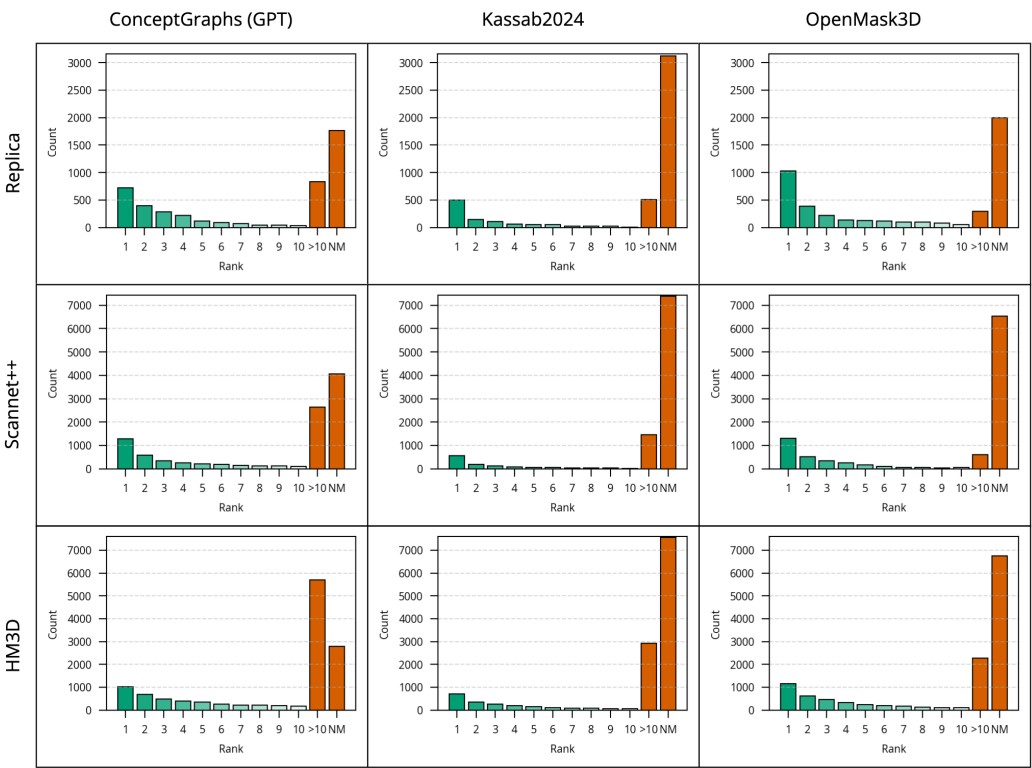

Figure 15: **Query Rank Counts.** We embed each OpenLex3D query and rank the predicted objects of each method based on feature similarity. If the target ground truth instance has an IoU of at least 0.25 with a predicted instance, we report the corresponding predicted rank. Otherwise, the query counts as "No Match" (NM). Queries with a rank of 10 or above are grouped together. The high number of unmatched queries confirms that most methods still struggle with the geometric aspect of the task and fail to segment instances that sufficiently overlap with the ground truth.