# OpenReview forum: "OpenLex3D: A Tiered Benchmark for Open-Vocabulary 3D Scene Representations"
_NeurIPS.cc/2025/Datasets_and_Benchmarks_Track — NeurIPS 2025 Datasets and Benchmarks Track poster_

### Official Review · Reviewer_oPAW · 2025-06-24

**Rating:** 5
**Confidence:** 4

**Summary:**

The paper introduces OpenLex3D, a novel benchmark designed to evaluate open-vocabulary 3D scene representations. Traditional benchmarks rely on closed-set semantics, limiting their ability to capture the richness and variability of natural language. OpenLex3D addresses this gap by providing a tiered evaluation framework with human-annotated labels for scenes from three datasets: Replica, ScanNet++, and HM3D. The benchmark includes synonyms, depictions, visually similar objects, and clutter as label categories, offering 13 times more labels per scene than original datasets. The authors propose two evaluation tasks—tiered semantic segmentation and object retrieval—and assess several state-of-the-art methods, revealing insights into their strengths and weaknesses.

**Dataset Code Accessibility:**

Yes

**Dataset Code Comments:**

The proposed contributions of the work are accessible:
1: Annotations: The annotation files can be downloaded via AISDatasets.
2: Tutorial: Instructions for integrating the data with ScanNet and HM3D are clear and easy to follow.
3: Evaluation: The evaluation code and standardized format are well-documented and easy to understand.

**Ethical Considerations:**

No, there are no or only very minor ethics concerns

**Final Justification:**

I thank again for the detailed rebuttal. Many of my concerns have been addressed, and I appreciate the clarifications. However, some potential issues remain:

The number of scenes is still limited. While the labeling limitations are understandable given the associated costs, there is potential for improvement in future work..

The annotations relyon the masks from the original dataset, which are relatively coarse and may miss small objects. However, this is understandable since it is the best available public dataset.

I will maintain my current rating.

**Limitations Weaknesses:**

1: Table 5 in the supplementary provides details of each scene; however, compared to semantic segmentation benchmarks like ScanNet++ (50 scenes) and HM3D (370 scenes), selecting only 10 scenes per task may limit cross-scene variability. Would it be possible to report performance variance across the 10 selected scenes to better understand scene-level robustness?
2: Overall, Table 1 suggests there are relatively few ambiguous objects. To better understand the differences, could you provide examples of ambiguous cases from ScanNet++ and HM3D, particularly illustrating why HM3D exhibits more ambiguity?
3: The clutter category is defined based on IoU > 0, but the rationale behind this design choice is unclear. Would it be more meaningful to incorporate a weighting scheme that accounts for the distance to neighboring objects? For example, in Figure 4 (Replica), a thin strip in the window frame is labeled as clutter—this type of clutter arguably deserves a smaller penalty compared to larger misclassifications, such as labeling an entire sofa as clutter.
4: Labelling bias: Using synonyms, descriptions, and cluttered labeling aligns very well with open-vocabulary tasks, as reflected in the label distribution results shown in Section E. However, the textual annotations was generated by annotators from different backgrounds and professions. Given that only four annotators were involved, could there be a risk of bias?
5: Small objects:Since the method adopts an instance-level object segmentation approach similar to OpenMask3D, is there a possibility of missing small objects (e.g., items on a bookshelf or within shelves)? This lack of fine-grained detail might contribute to the increased use of synonyms.
6: Missing object-object, object-scene description: Datasets like MMScan and Mosaic3D explore spatial relationships between objects and their surrounding scenes, making them more suitable for real-world applications. However, such discussions appear to be missing in this dataset.

**Strengths Contributions:**

Contributions:

1. **New Labeling Scheme**: Introduces a hierarchical labeling system with four categories (synonyms, depictions, visually similar, clutter) to capture real-world linguistic variability.
2. **Dataset Expansion**: Provides **3,812 newly labeled objects** across three datasets, significantly expanding annotation diversity.
3. **Evaluation Tasks**: Proposes **tiered semantic segmentation** and **object retrieval** tasks with novel metrics (Top-N Frequency, Set Ranking) to assess open-vocabulary methods.
4. **Public Benchmark**: Releases the OpenLex3D toolkit and labeled data publicly for community use.
5: **Comprenhensive experiments and analysis**: The work presents a comprehensive set of experiments with in-depth analysis that is clear and easy to follow.

---

> ### Author Rebuttal · Authors · 2025-07-30
>
> We thank the reviewer for their insightful feedback and appreciate that they find our analysis “clear and easy to follow” and that our labels significantly expand “annotation diversity”. We address each of their comments below:
>
> **Variance results** *“1 [..] report performance variance across the 10 selected scenes to better understand scene-level robustness?”* We agree with the concern put forward and list an excerpt of the standard deviation of the Top-N frequency metric in the following three tables:
>
> Replica:
> | Method      | Synonyms | Depictions | Vis Sim | Clutter | Missing | Incorrect |
> |-------------|----------|------------|---------|---------|---------|-----------|
> | ConceptGraphs   | 0.14     | 0.02       | 0.07    | 0.08    | 0.01    | 0.11      |
> | ConceptFusion    | 0.07     | 0.01      | 0.05    | 0.03    | 0.01     | 0.09      |
> | OpenMask3D      | 0.08     | 0.02       | 0.10    | 0.11    | 0.07    | 0.05      |
>
> ScanNet++:
> | Method      | Synonyms | Depictions | Vis Sim | Clutter | Missing | Incorrect |
> |----------------|---------------|------------|---------|---------|---------|-----------|
> | ConceptGraphs   | 0.04     | 0.02       | 0.03    | 0.05    | 0.05    | 0.07      |
> | ConceptFusion    | 0.07     | 0.01      | 0.01    | 0.03    | 0.02    | 0.08      |
> | OpenMask3D    | 0.10     | 0.01       | 0.02    | 0.13    | 0.05    | 0.11      |
>
> HM3D:
> | Method      | Synonyms | Depictions | Vis Sim | Clutter | Missing | Incorrect |
> |-------------|----------|------------|---------|---------|---------|-----------|
> | ConceptGraphs   | 0.07     | 0.02       | 0.03    | 0.03    | 0.02    | 0.07      |
> | ConceptFusion    | 0.02     | 0.01      | 0.01    | 0.01   | 0.05    | 0.07      |
> | OpenMask3D    | 0.05   | 0.01       | 0.01    | 0.03    | 0.03    | 0.04      |
>
> These tables show that the variance across scenes is non-negligible. For example, there is high variance in the clutter score for OpenMask3D. OpenMask3D relies on a fixed-class 3D segmentation model, Mask3D, and is therefore influenced by the objects which appear in the original training sets and may not generalize to out-of -distribution scenes. We will include full details of performance variance for each method and metric in the appendix and would like to thank you for bringing up this point.
>
> **Object ambiguity** *“2 [..] there are relatively few ambiguous objects”*: Objects were flagged as ambiguous when there was significant disagreement between annotators. HM3D contains more ambiguous cases than ScanNet++, because HM3D uses RGB-D images rendered from reconstructed scenes. The reconstruction process can lead to incomplete or distorted representations of objects, making it harder for our labellers to label or recognize objects compared to the real RGB-D captures in ScanNet++. While we are unable to include visual examples in this rebuttal, we will provide illustrative examples in the appendix of our paper.
>
> **Clutter category** *“3 [..] more meaningful to incorporate a weighting scheme that accounts for the distance to neighboring objects?”*: We selected an IoU>0 threshold for the clutter category to ensure that any neighboring objects overlapping with a target object were included. The idea of incorporating a weighting scheme is interesting. However, such a weighting system would be inherently subjective and highly dependent on the specific objects and their spatial context. We feel that it is unclear how a weighting could be objectively integrated into the current metric. Nonetheless, it is still an interesting idea to explore, and we thank the reviewer for their suggestion.
>
> **Labeling bias** *"4 [..] only four annotators were involved, could there be a risk of bias?”* We acknowledge that a certain degree of annotator bias is inevitable. To mitigate this, we aimed to ensure diversity among our labelers. They come from a range of English-speaking backgrounds, including Canada, India, the UK, and Europe. This diversity helps to capture subtle linguistic nuances. Additionally, we made an effort to balance labelling and reviewing assignments so that each object was annotated or reviewed by labelers from different backgrounds. All these measures sought to achieve a more consistent and representative distribution across the dataset.
>
> **Object masks** *“[..] adopts an instance-level object segmentation approach similar to OpenMask3D [..]”*: We would like to clarify that the object segmentations used in our benchmark are the ground-truth masks provided by the original datasets. While the level of granularity in these masks varies slightly across datasets, they do include many fine-grained examples (for example small objects on shelves) demonstrating reasonable coverage of detailed scene elements.
>
> **Edge labels** *“6: Missing object-object, object-scene description [..]”*:  We acknowledge that a limitation of our benchmark is the exclusion of object-object and object-scene descriptions. We chose not to include these, as object-object relationships are often primarily spatial in nature and are, as such, not subject to semantic ambiguity, which we study with our proposed category definitions. In terms of semantic object-object relations and free-text labeling of those, there is, to date, no strict evaluation protocol that does not rely on large language models, which may have been used to produce predictions in the first place. Nonetheless, we agree with this concern and will expand the discussion in our Limitations section.

---

> > ### Comment · Reviewer_oPAW · 2025-08-03
> >
> > Thank you for the detailed response. Many of my concerns have been addressed, and I appreciate the clarifications. However, some potential issues remain:
> >
> > - The number of scenes is still limited. While the labeling limitations are understandable given the associated costs, there is potential for improvement in future work..
> >
> > - The annotations relyon the masks from the original dataset, which are relatively coarse and may miss small objects. However, this is understandable since it is the best available public dataset.
> >
> > I will maintain my current rating.

---

### Official Review · Reviewer_4Vtt · 2025-06-25

**Rating:** 5
**Confidence:** 4

**Summary:**

This work introduces OpenLex3D, a dedicated benchmark for evaluating 3D open-vocabulary scene understanding. It provides enriched annotations for scenes from Replica, ScanNet++, and HM3D, covering diverse relationships such as synonyms, depictions, visual similarity, and clutter. The benchmark defines two tasks—open-set 3D semantic segmentation and object retrieval—and conducts comprehensive evaluations of existing 3D open-vocabulary methods, highlighting their respective strengths and limitations. Extensive experiments are conducted, and the code is publicly available.

**Dataset Code Accessibility:**

Yes

**Ethical Considerations:**

No, there are no or only very minor ethics concerns

**Final Justification:**

This work provides enriched annotations for scenes from Replica, ScanNet++, and HM3D, covering diverse relationships such as synonyms, depictions, visual similarity, and clutter. These contributions are important for advancing current 3D scene understanding tasks. I hold a positive attitude toward this work.

**Limitations Weaknesses:**

1. As mentioned in the limitations, the current annotations lack finer-grained attributes such as **affordances**, **material**, and **color**, which could further enhance the benchmark’s utility and realism.

2. It would be interesting to explore whether **fine-tuning GPT-based methods** on OpenLex3D could improve their ability to distinguish between the four challenging categories—**synonyms**, **depictions**, **visual similarity**, and **clutter**—and thereby lead to more accurate open-vocabulary classification.

3. The **Related Work** section is relatively limited and could benefit from a broader discussion of recent advances, including:

   * *Open-Vocabulary 3D Semantic Segmentation with Foundation Models*
   * *SAS: Segment Any 3D Scene with Integrated 2D Priors*
   * *Mosaic3D: Foundation Dataset and Model for Open-Vocabulary 3D Segmentation*
   * *OpenIns3D: Snap and Lookup for 3D Open-vocabulary Instance Segmentation*
   * *......*

**Strengths Contributions:**

1. The proposed categories—synonyms, depictions, visual similarity, and clutter—are insightful and help reveal the limitations of existing methods in open-vocabulary 3D understanding.

2. The authors conduct extensive experiments on both open-set 3D semantic segmentation and object retrieval tasks, with evaluations at both the semantic and instance levels.

3. The paper is relatively well-written and presents its ideas clearly.

---

> ### Author Rebuttal · Authors · 2025-07-30
>
> We thank the reviewer for their feedback and appreciate that they found our proposed categories insightful and that our ideas were clearly presented. We respond to each of their concerns below:
>
> **Finer-grained attributes** *“1. [..] current annotations lack finer-grained attributes such as affordances, material, and color [..]”*: We recognize the value of extending our annotations to finer-grained attributes such as affordances, materials, and colors. Our current categories are primarily designed to capture different levels of specificity, rather than object-level attributes. Expanding in this direction could provide a valuable avenue for future work, enabling a separate evaluation of how much fine-grained information current methods can extract per object. We also note that other works, such as OpenScan, specifically target this aspect of scene understanding.
>
> **Fine-tuning GPT-based methods** *“2. [..] whether fine-tuning GPT-based methods on OpenLex3D could improve their ability to distinguish between the four challenging categories [..]”*: The primary aim of this work is on improving evaluation methods for zero-shot performance, and as such, we did not focus on training or fine-tuning models. Nonetheless, we value the suggestion and agree that exploring the use of our categories for fine-tuning current methods, would be an interesting direction for future work. This could use contrastive or in-context learning.
>
> *“The Related Work section is relatively limited [..]”*: We thank the reviewer for highlighting these relevant works. We will include them in the related work section of our manuscript.

---

> > ### Comment · Reviewer_4Vtt · 2025-08-04
> >
> > Thank you for the detailed response. This is a solid piece of work that has the potential to improve existing 3D scene understanding methods. I appreciate the clarifications provided and update my score to Accept.

---

### Official Review · Reviewer_t9s4 · 2025-06-29

**Rating:** 5
**Confidence:** 3

**Summary:**

The paper introduces OpenLex3D, a new benchmark aimed at evaluating open-vocabulary 3-D scene representations.

- Scope & Data: 3 widely–used indoor RGB-D datasets (Replica, ScanNet++, HM3D) are re-annotated with free-form text labels, expanding each scene’s label set by ≈13×.

- Tiered Label Taxonomy: Four hierarchical categories of semantic specificity, Synonyms, Depictions, Visually Similar, and Clutter, allow systematic diagnosis of different misclassification modes.

- Tasks & Metrics : Two evaluation tasks are defined:
Tiered open-set semantic segmentation with new Top-N Frequency and Set Ranking metrics and
open-set object retrieval with 200–1 500 auto-generated queries per scene.

- Baselines: 4 state-of-the-art object-centric and 2 dense methods are benchmarked. Results show no single approach excels across both tasks, highlighting challenges in feature fusion and segmentation.

**Dataset Code Accessibility:**

Yes

**Dataset Code Comments:**

The author provides the GitHub link of this work.

**Ethical Considerations:**

No, there are no or only very minor ethics concerns

**Final Justification:**

My concerns are addressed and I will keep my rating.

**Limitations Weaknesses:**

There are still some limitations of this work:
- Because the authors reuse the original instance masks, a “sofa” is still a single blob; cushions, armrests, and handles are not labeled separately. Fine-grained part reasoning is therefore out of scope.
- Retrieval questions are auto-generated with a fixed phrase pattern, so they don’t reflect the full linguistic variety one would expect from real users.
- The overall amount of data is still very limited. There are only 23 scenes in total across all three datasets.

**Strengths Contributions:**

The strengths of this work are:
- The community has lacked a benchmark that truly tests open-vocabulary 3-D scene understanding. OpenLex3D squarely fills that hole, so its practical value is immediately obvious.

- Tiered label taxonomy shifts focus from “exact match” to graded correctness. Instead of the usual all-or-nothing label match, the four-level taxonomy (Synonyms → Depictions → Visually-Similar → Clutter) lets researchers pinpoint why a prediction failed, not just that it failed.

- The data quality is high. Each of the 3812 objects is annotated by four different people and then cleaned through a structured post-processing step, giving the dataset a level of polish that many existing benchmarks lack.

- The narrative flows well, formulas are concise, and figures/tables are color-coded and easy to digest.

---

> ### Author Rebuttal · Authors · 2025-07-30
>
> We thank the reviewer for their feedback and are pleased that the reviewer appreciates the practical value of the benchmark and the quality of the data. In the following we respond to each of the concerns:
>
> **Original instance segmentation** *“ [..] the authors reuse the original instance masks, a “sofa” is still a single blob; cushions, armrests, and handles are not labeled separately. Fine-grained part reasoning is therefore out of scope [..]”*: We acknowledge that using the original instance masks carries forward the limitations of the source datasets. However, our primary goal is to provide diverse labels at the object level, where such diversity is most impactful. In contrast, object parts typically exhibit less label variety. Although the original instance masks may omit certain parts (such as handles) they do capture fine-grained objects like individual books, cushions, and other small items. Preserving these original segments also ensures backward compatibility. We note that other works, such as SceneFun3D, are specifically designed to address part-level annotations.
>
> **Fixed phrase pattern for retrieval** *“Retrieval questions [..] don’t reflect the full linguistic variety one would expect from real users”:* We acknowledge that the retrieval questions follow a fixed-phrase pattern. This approach was chosen as a simple way to generate a large number of queries per scene. We note that even with the current templated queries and diverse object labels, the task remains challenging for existing methods, as shown in Table 4. However, as reviewer t9s4 suggests, we agree that generating more natural phrasings (e.g. using GPT) would be an interesting direction for future work.
>
> **Overall amount of data** *“only 23 scenes in total across all three datasets”*:  We acknowledge that the total number of labeled scenes in our benchmark is small with respect to the overall original datasets, however, each scene is labeled in a much more detailed manner. The size of OpenLex3D roughly reflects current practices in published papers, where typically 8-10 scenes are selected from each dataset for evaluation. We aimed for a trade-off between label variety per object and total number of objects. That said, we recognize the impact of further expanding the number of labeled scenes and can imagine scaling up in future iterations. Potentially, this could also include crowd-sourced labels under strict label cross-verification.

---

### Official Review · Reviewer_7gzP · 2025-07-02

**Rating:** 5
**Confidence:** 3

**Summary:**

This manuscript presents a novel benchmark OpenLex3D to evaluate 3D open-vocabulary scene understanding and representations. OpenLex3D is based on three prominent RGB-D datasets: Replica, Scannet++ and HM3D, and provides tiered label annotations. Each object is hand-annotated with four label categories: synonyms, depictions, visually similar, and clutter. This manuscript also introduces two open-set 3D scene understanding tasks and evaluation metrics. Extensive experiments offer great insight for open-vocabulary 3D scene understanding performance and future improvements.

**Dataset Code Accessibility:**

Yes

**Ethical Considerations:**

No, there are no or only very minor ethics concerns

**Final Justification:**

All my concerns have been illustrated.

**Limitations Weaknesses:**

While this is a strong submission and a valuable contribution, it is unclear how the proposed benchmark can facilitate improvement in future open-vocabulary 3D scene understanding methods. It would be interesting to discuss the implications of this benchmark.

**Strengths Contributions:**

1. The proposed benchmark OpenLex3D is well-suited for evaluating open-vocabulary 3D scene understanding. The hand-annotated, tiered label structure enables comprehensive evaluation of model performance in predicting both correct and incorrect labels. This addresses a critical issue in existing 3D scene understanding works, where no open-set semantic segmentation labels and metrics are available.
2. The open-set 3D semantic segmentation and object retrieval task introduced in the manuscript is well defined. The evaluation of four state-of-the-art open-vocabulary 3D scene understanding frameworks offers valuable insights into the strengths and limitations of existing approaches, while also demonstrating the utility of the proposed benchmark.
3. The manuscript provides detailed benchmark statistics, including label distribution and labeling process.

---

> ### Author Rebuttal · Authors · 2025-07-30
>
> We thank the reviewer for their feedback and we are pleased that they agree that our benchmark could address a critical issue in 3D scene understanding and offer diagnostic insights.
>
> *[..] it is unclear how the proposed benchmark can facilitate improvement in future open-vocabulary 3D scene understanding methods [..]:*
> Our primary goal with this work is to provide a benchmark that enables users to systematically evaluate their models and to identify distinct failure modes. For instance, if a system yields a high score on the clutter metric, this would indicate specific weaknesses in the system’s instance segmentation module. Existing evaluation systems lack this level of detailed feedback. This level of interpretability helps to guide targeted improvements to open-vocabulary scene understanding systems.

---

### Decision · Program_Chairs · 2025-09-18

**Decision:**

Accept (poster)

**Comment:**

The paper proposes a benchmark on 3D open-vocabulary scene understanding. The work appears to be technically sound and to make a contribution in a topic of relevance to NeurIPS.

The initial reviews identified some weaknesses and concerns, with key issues about: lack of clarity or insights on how future open-vocabulary 3D scene understanding methods can improve, limited dataset scale, and concerns about evaluation implementation details involving granularity of semantic segmentation and categorization based on segmentation masks from prior datasets.

After the rebuttal and discussion, all four reviewers found their concerns to be sufficiently addressed and recommend acceptance.  The AC did not find a basis to overrule the reviewer opinion.  The AC recommends to incorporate clarifications and improvements from the rebuttal and discussion into the camera ready version of the paper.

===== FINAL UPDATE FROM DB Track PCs ====

The final decision for this paper has been taken by the program chairs after consultation with the SACs. All Senior Area Chairs have ranked papers according to the feedback from the AC during the review process. We decided to leave the original meta-review to reflect the opinion of the AC in light of the initial discussions with reviewers and SAC.